# Shear stimulation of FOXC1 and FOXC2 differentially regulates cytoskeletal activity during lymphatic valve maturation

**Pieter R Norden[1†], Amélie Sabine[2†], Ying Wang[3], Cansaran Saygili Demir[2], Ting Liu[1], Tatiana V Petrova[2], Tsutomu Kume[1]\***

[1]Feinberg Cardiovascular and Renal Research Institute, Department of Medicine, Feinberg School of Medicine, Northwestern University, Chicago, United States; [2]Department of Oncology, Centre Hospitalier Universitaire Vaudois and University of Lausanne, Epalinges, Switzerland; [3]Department of Biochemistry and Molecular Biology, College of Medicine and Science, Mayo Clinic, Jacksonville , United States

**Abstract** Mutations in the transcription factor *FOXC2* are predominately associated with lymphedema. Herein, we demonstrate a key role for related factor FOXC1, in addition to FOXC2, in regulating cytoskeletal activity in lymphatic valves. FOXC1 is induced by laminar, but not oscillatory, shear and inducible, endothelial-specific deletion impaired postnatal lymphatic valve maturation in mice. However, deletion of *Foxc2* induced valve degeneration, which is exacerbated in *Foxc1; Foxc2* mutants. *FOXC1* knockdown (KD) in human lymphatic endothelial cells increased focal adhesions and actin stress fibers whereas *FOXC2*-KD increased focal adherens and disrupted cell junctions, mediated by increased ROCK activation. ROCK inhibition rescued cytoskeletal or junctional integrity changes induced by inactivation of FOXC1 and FOXC2 *invitro* and *vivo* respectively, but only ameliorated valve degeneration in *Foxc2* mutants. These results identify both FOXC1 and FOXC2 as mediators of mechanotransduction in the postnatal lymphatic vasculature and posit cytoskeletal signaling as a therapeutic target in lymphatic pathologies.

**\*For correspondence:**
t-kume@northwestern.edu

†These authors contributed equally to this work

**Competing interests:** The authors declare that no competing interests exist.

## Introduction

The lymphatic vasculature has a critical role in maintaining tissue homeostasis by returning interstitial fluid to the venous circulation, absorbing lipids from the digestive tract, and providing a network for immune surveillance and response (*Adams and Alitalo, 2007*; *Zheng et al., 2014*; *Norden and Kume, 2020*). Mutations identified in genes involved in the VEGF-C/VEGFR3 signaling pathway are commonly associated with primary lymphedema and other lymphatic malformations, which include mutations in critical transcription factors such as GATA2 and FOXC2 (*Brouillard et al., 2014*; *Ostergaard et al., 2011*; *Kazenwadel et al., 2012*; *Kazenwadel et al., 2015*; *Bell et al., 2001*; *Fang et al., 2000*; *Finegold et al., 2001*; *Dagenais et al., 2004*; *van Steensel et al., 2009*; *Fauret et al., 2010*; *Witte et al., 2009*). During murine embryonic development at E15.5, the primitive lymphatic vascular plexus remodels and is reorganized into capillaries, pre-collectors and collecting vessels characterized by differential protein expression patterns, cell-cell junctions, and mural cell recruitment (*Schulte-Merker et al., 2011*). Coinciding with the remodeling of this plexus is the formation of intraluminal, bi-leaflet valves in the collecting vessels that function to prevent lymph backflow (*Bazigou and Makinen, 2013*). Critical to lymphatic valve formation, FOXC2 and GATA2 are upregulated in lymphatic endothelial cells (LECs) in response to oscillatory shear stress (OSS) forces (*Kazenwadel et al., 2015*; *Sabine et al., 2012*). In valve forming cells that express high levels of the master lymphatic regulator PROX1, FOXC2 cooperates with PROX1 to control intraluminal

invagination of LECs and reorganization into valve forming leaflets that form mature structures by postnatal day (P)1 (*Kazenwadel et al., 2015*).

FOXC1 and FOXC2 are closely related members of the forkhead box (FOX) transcription factor family with nearly identical DNA binding domains, similar expression patterns in mesenchymal tissues during development, and essential roles in cardiovascular developmental processes (*Kume et al., 2001*; *Seo et al., 2006*; *Seo and Kume, 2006*; *Kume, 2009*). Mutations in human *FOXC1* have primarily been dominantly associated with eye anterior segment defects, cerebellar malformation, and cerebral small vessel disease. In contrast, mutations in *FOXC2* have been dominantly associated with lymphedema-distichiasis syndrome characterized by failure of lymph drainage in limbs, venous valve failure, and the growth of an extra set of eyelashes (*Tümer and Bach-Holm, 2009*; *Micheal et al., 2016*; *Aldinger et al., 2009*; *French et al., 2014*; *Fang et al., 2000*; *Traboulsi et al., 2002*; *Tavian et al., 2016*; *Mellor et al., 2007*). Work from our group has demonstrated that during lymphatic collecting vessel maturation and valve formation, FOXC2 regulates connexin 37 expression and activation of calcineurin/NFAT signaling (*Petrova et al., 2004*; *Norrmén et al., 2009*; *Sabine et al., 2012*). Additionally, FOXC2 was shown to be crucial for lymphatic valve maintenance by regulating LEC junctional integrity and cellular quiescence under reversing flow conditions via restriction of TAZ-mediated proliferation (*Sabine et al., 2015*). Furthermore, our group also demonstrated that FOXC1 and FOXC2 negatively regulate increased Ras/ERK signaling during embryonic lymphangiogenesis to suppress formation of hyperplastic lymphatic vessels, which are also observed in individuals with *FOXC2* mutations (*Fatima et al., 2016*; *Brice et al., 2002*; *Mansour et al., 1993*). However, while a critical role for FOXC2 has been established during postnatal valve formation and maturation, the role of FOXC1, and potentially cooperative role of both transcription factors, is poorly understood.

Here, we report an essential role for FOXC1 during lymphatic valve maturation and maintenance. Detailed comparison of FOXC1 and FOXC2 expression and roles in lymphatic valves suggests some overlap with a broader importance for FOXC2, but more subtle, key contribution for FOXC1. In mice, endothelial cell (EC)-specific deletion of *Foxc1* postnatally impairs valve maturation, while *Foxc2* deletion impairs maturation and induces valve degeneration, as previously described (*Sabine et al., 2015*). However, combined deletion of *Foxc1* and *Foxc2* worsens the phenotype induced by single deletion of *Foxc2*. In mature lymphatic valves FOXC1 is expressed at high levels in a subset of LECs that also express high levels of FOXC2 at the rim of valve leaflets. This specific pattern of FOXC1 expression is likely due to its upregulation, like FOXC2, by unidirectional shear, while only FOXC2 is upregulated by OSS. *In vitro* loss of FOXC1 or FOXC2 induced hyper-activation of contractile stress fibers in LECs; however, a striking difference is their association with focal adhesions upon *FOXC1* knockdown *vs* focal adherens junctions upon *FOXC2* knockdown. This phenotype is rescued by inhibition of Rho-associated protein kinase (ROCK) *in vitro*, which also improves LEC barrier integrity *in vivo*, while valve degeneration is partially rescued in only *Foxc2* mutants. Finally, via generation of transgenic mice that express *Foxc2* within the *Foxc1* locus, we show *Foxc2* is capable of functionally substituting for *Foxc1* in lymphatic development and maturation. Together, our data show a complementary role for FOXC1 in addition to FOXC2 as key mediators of mechanotransduction in the postnatal lymphatic valves and implicate new mechanistic targets for therapeutics in the treatment of lymphatic-associated diseases.

## Results

### FOXC1 and FOXC2 are required for postnatal lymphatic valve maturation and maintenance

Our group previously reported that FOXC1 and FOXC2 expression co-localizes with PROX1 in lymphatic valve-forming cells at E17 and later at P3 (*Fatima et al., 2016*). However, the expression pattern of FOXC1 in the mesenteric lymphatic collecting vessels and valves in adult mice remains unknown. We first characterized the expression pattern of FOXC1 and FOXC2 in mature valves of 4 week old adult mice to delineate possible differential or cooperative roles during valve maturation and maintenance. Immunostaining of mesentery tissue with FOXC1, FOXC2, and VEGFR3 antibodies identified colocalization of FOXC1 and FOXC2 within the nuclei of intraluminal valve leaflets while FOXC2 expression was more highly enriched in the valve sinuses and surrounding lymphangion

compared to FOXC1 (*Figure 1*). Of note, FOXC1 expression was most highly enriched in cells located at the leading free-edge (*Bazigou et al., 2009*; *Danussi et al., 2013*; *Bazigou and Makinen, 2013*; *Sabine et al., 2018*) of the intraluminal side of valve leaflets exposed to pulsatile laminar shear stress (LSS) forces during valve opening/closure cycles (*Sabine et al., 2016*).

Previous work from our group has demonstrated the critical role for FOXC2 in postnatal lymphatic vascular function (*Sabine et al., 2015*) and we have also shown that EC-specific deletion of *Foxc1* during murine embryonic development impairs lymphatic valve maturation (*Fatima et al., 2016*). However, the temporal regulatory role of FOXC1 transcriptional activity during valve development, maturation, and maintenance is not well understood. To investigate FOXC1 function in the early postnatal lymphatic vasculature and valve regions, we crossed conditional null *Foxc1^{fl}* mice (*Sasman et al., 2012*) with *Chd5-Cre^{ERT2}* mice (*Sörensen et al., 2009*) to generate tamoxifen-inducible, EC-specific *Foxc1* mutant (EC-*Foxc1*-KO) mice. Tamoxifen was administered from P1-P5 to induce *Cre*-mediated recombination and we confirmed deletion of *Foxc1* via qPCR analysis of isolated CD31-positive cells from hearts of P6 individuals and by immunostaining of the mesenteric lymphatic vasculature with antibodies against FOXC1, FOXC2, and VEGFR-3 (*Figure 2—figure supplement 1a,d,e*). We next investigated whether lymphatic valve maturation and maintenance

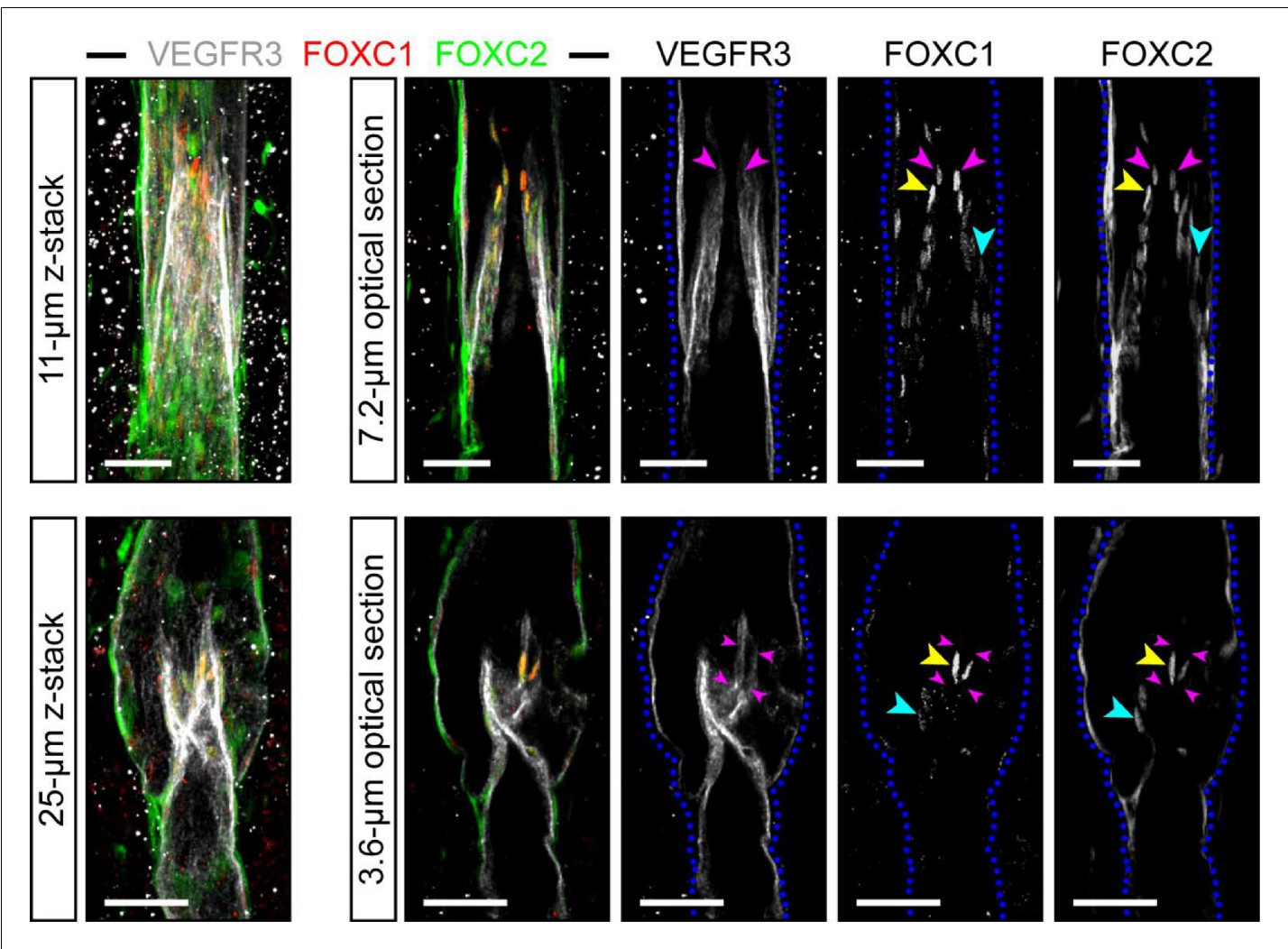

**Figure 1.** FOXC1 is highly expressed in a subset of LECs at the free edge of lymphatic valve leaflets. Representative images of maximum intensity projections (left) and optical sections (right) from mesentery collecting vessels of a 4 week old C57Bl6 mouse immunostained with VEGFR3 (white), FOXC1 (red), and FOXC2 (green). Purple arrowheads denote the position of valve leaflet free-edges. Yellow arrowheads denote FOXC1^{HIGH}/FOXC2^{HIGH} LECs located near the leaflet free-edge. Blue arrowheads denote FOXC2-positive LECs in valve leaflets with only weakly expressed FOXC1. Dashed blue lines on the single-channel images outline the vessel borders. Scale bars are 50 μm.

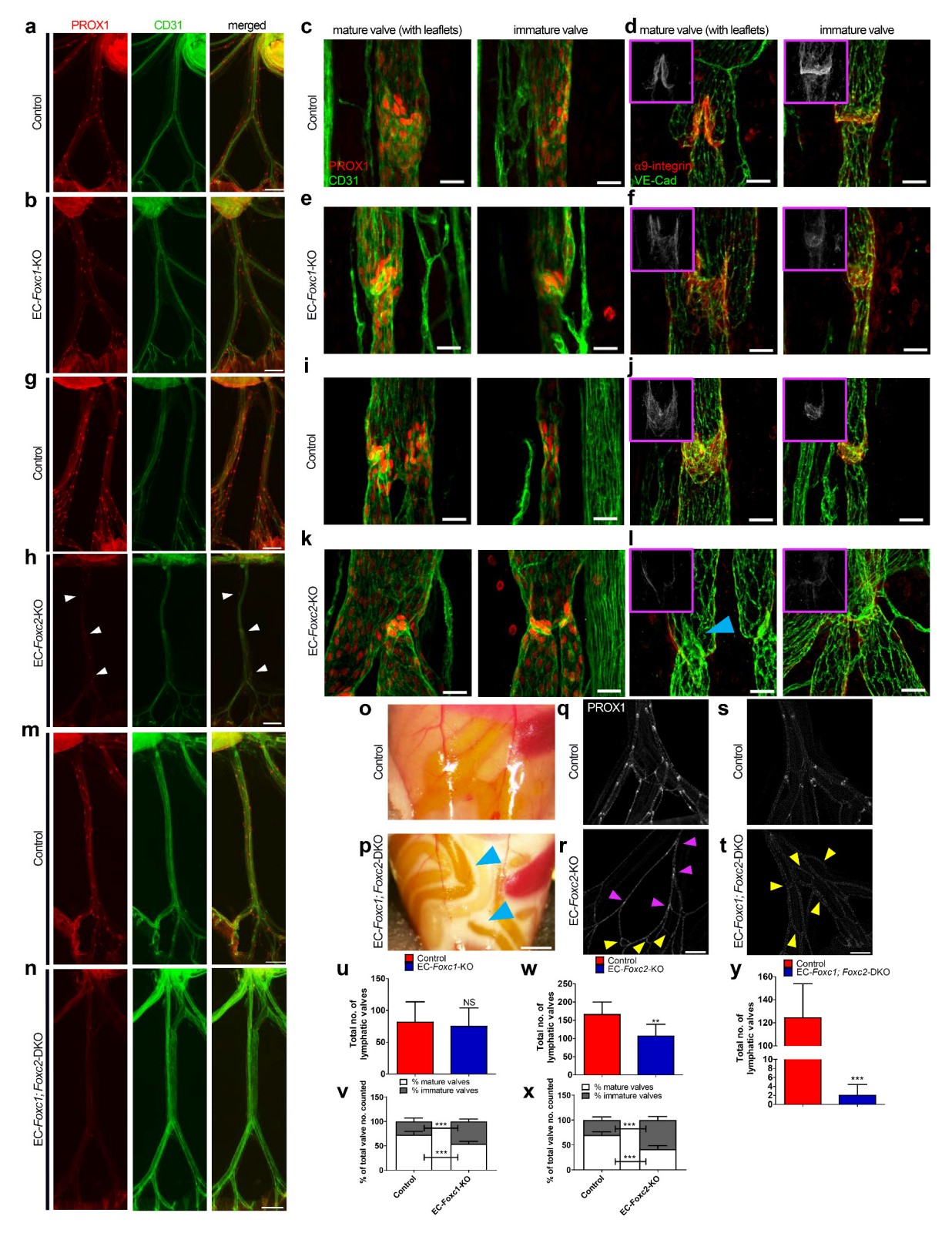

**Figure 2.** Compound endothelial-specific *Foxc1; Foxc2* mutants present severe chylous ascites and are nearly absent of PROX1-high expressing lymphatic valves. (a, b, g, h, m, n) Representative images of PROX1 and CD31 immunostained lymphatic collecting vessels in P6 littermate control (a, g, m) and EC-*Foxc1*-KO (b), EC-*Foxc2*-KO (h), and EC-*Foxc1; Foxc2*-DKO (n) individuals. White arrowheads denote PROX1-high valves. Scale bars are 500 μm. (c – f, i – l) Representative images of mature and immature lymphatic valves immunostained with PROX1 and CD31 or α9-integrin and VE-Cadherin

*Figure 2 continued on next page*

*Figure 2 continued*

in P6 littermate control (c, d, i, j) and EC-*Foxc1*-KO (e, f) or EC-*Foxc2*-KO individuals (k, l). Pink inserts denote single channel α9-integrin (white) images. Blue arrowhead denotes VE-Caderhin-positive intraluminal valve leaflet with markedly reduced α9-integrin expression in l. Scale bars are 25 μm. (o, p) Appearance of chylous ascites in the peritoneal cavity of a P6 *EC-Foxc1; Foxc2*-DKO mouse (p) compared to littermate control (o). Blue arrow heads indicate chylous effusion; scale bar equals 1 mm. (q – t) Representative images of PROX1 immunostained lymphatic collecting vessels in P6 littermate control (q, s) and EC-*Foxc2*-KO (r) or *EC-Foxc1; Foxc2*-DKO (t) individuals show degeneration of lymphatic valves in *Foxc2* mutants and regression of collecting vessels into a primitive lymphatic architecture in EC-specific *Foxc1* and *Foxc2* mutants. Pink arrowheads denote degenerating PROX1-high expressing valve regions. Yellow arrowheads highlight looping and interconnections between branches of collecting vessels. Scale bar is equal to 200 μm. (u) Quantification of total lymphatic valve number (identified by PROX1-high expression) in lymphatic collecting vessels of P6 Control and EC-*Foxc1*-KO individuals. N = 7 for Control and N = 7 for EC-*Foxc1*-KO individuals, four collecting vessels analyzed per individual. (v) Percentage of mature and immature lymphatic valves normalized to total valves counted in P6 Control and EC-*Foxc1*-KO individuals. (w) Quantification of total lymphatic valve number in lymphatic collecting vessels of P6 Control and EC-*Foxc2*-KO individuals. N = 9 for Control and N = 9 for EC-*Foxc2*-KO individuals, four collecting vessels analyzed per individual. (x) Percentage of mature and immature lymphatic valves normalized to total valves counted in P6 Control and EC-*Foxc2*-KO individuals. (y) Quantification of total lymphatic valve number in lymphatic collecting vessels of P6 Control and *EC-Foxc1; Foxc2*-DKO individuals. N = 6 for Control and N = 6 for *EC-Foxc1; Foxc2*-DKO individuals, four collecting vessels analyzed per control individual and all lymphatic collecting vessels assessed per mutant individual. Data are presented as mean (± SD) and analyzed using Student's t-test. NS denotes no significance. ** denotes p<0.01. *** denotes p<0.001.

The online version of this article includes the following figure supplement(s) for figure 2:

**Figure supplement 1.** Expression of *Foxc1* and *Foxc2* mRNA and FOXC1 and FOXC2 protein levels in endothelial-specific knockout models.

**Figure supplement 2.** LEC-specific knockout of *Foxc1* recapitulates phenotype observed in *Cdh5-Cre*^ERT2^; *Foxc1*^fl/fl^ knockout mice.

**Figure supplement 3.** α9-integrin expression is absent in lymphatic collecting vessels of EC-specific *Foxc1; Foxc2* mutant mice.

**Figure supplement 4.** Cell elongation and junctional integrity is markedly impaired in compound EC-specific *Foxc1; Foxc2* mutants, accompanied by increased apoptosis in lymphatic collecting vessels.

was affected in collecting vessels after induction of Cre-mediated recombination postnatally (*Figure 2a–f,u,v*). Quantification of regions with PROX1-high expression within lymphatic collecting vessels, indicative of valves, showed there was not a significant difference in total valve number in mice with inactivated *Foxc1* compared to littermate controls at P6 (*Figure 2u*). As VE-Cadherin, coded by the *Cdh5* gene, is also expressed in the blood vasculature, we immunostained the mesenteric vasculature with CD31 antibody and found no obvious changes associated with loss of *Foxc1* (*Figure 2a,b*). Using EC-specific *Foxc1* mutant mice generated from the *Tek*-Cre strain, our group previously reported that the proportion of lymphatic valves that had formed mature, v-shaped or semilunar bi-leaflet structures at P7 was significantly reduced in EC-specific *Foxc1* mutant mice compared to littermate controls (*Fatima et al., 2016*). To investigate whether this result was recapitulated using the inducible *Cdh5-Cre* strain, we also performed immunostaining of mesentery tissue for expression of a marker of valve maturation, α9-integrin, which is a receptor that interacts with its extracellular matrix (ECM) protein ligand fibronectin-EIIIA to regulate the formation of the ECM core of valve leaflets (*Bazigou et al., 2009*) in addition to PROX1/CD31 immunostaining (*Figure 2c – f*). Quantification of the proportion of lymphatic valves that had formed mature, v-shaped or semilunar bi-leaflet structures showed a significant reduction in EC-*Foxc1*-KO mice (*Figure 2v*), thus recapitulating our previously reported results using the *Tek*-Cre strain. To verify that the phenotype we observed was not mediated by inactivation of *Foxc1* within the blood vasculature, we next crossed conditional null *Foxc1*^fl^ mice with *Prox1-Cre*^ERT2^ mice (*Srinivasan et al., 2007*) to generate inducible LEC-specific *Foxc1* mutant (LEC-*Foxc1*-KO) mice and administered tamoxifen postnatally. PROX1 immunostaining showed no significant differences in total valve number of LEC-*Foxc1*-KO mice (*Figure 2—figure supplement 2a,b,e*). Additionally, immunostaining of PROX1 and FOXC1 showed enrichment of FOXC1 expression in intraluminal valve leaflets of littermate control mice (*Figure 2—figure supplement 2c*) whereas mature lymphatic valves were still observed in LEC-*Foxc1*-KO mice with reduced FOXC1 expression (*Figure 2—figure supplement 2d*). However, quantification of mature lymphatic valve structures demonstrated that LEC-*Foxc1*-KO mice had significantly less mature valves (*Figure 2—figure supplement 2f*), recapitulating the phenotype observed utilizing the *Cdh5-Cre*^ERT2^ strain (*Figure 2*). While similar phenotypes were observed between both *Cdh5-Cre*^ERT2^- and *Tek-Cre- Foxc1* mutants, these data demonstrate that FOXC1 function is maintained during postnatal lymphatic valve maturation and maintenance as *Tek-Cre* is expressed embryonically as early as E7.5 (*Braren et al., 2006*).

Previously, our group reported inducible LEC-specific deletion of *Foxc2* via *Prox1-Cre^ERT2* (*Bazigou et al., 2011*) at P4 was shown to significantly reduce total valve number in mesenteric lymphatic collecting vessels and a subset of *Foxc2* mutants presented chylous ascites and chylothorax beginning as early as 3 days after tamoxifen treatment with eventual fully penetrant mortality (*Sabine et al., 2015*). To compare the roles of FOXC1 and FOXC2 in our model, we crossed *Cdh5-Cre^ERT2* mice with conditional null *Foxc2^fl* mice to generate EC-specific *Foxc2* mutant (EC-*Foxc2*-KO) mice and administered tamoxifen from P1-P5 with inactivation confirmed by qPCR in ECs isolated from P6 hearts and by immunostaining of the mesenteric lymphatic vasculature with antibodies against FOXC1, FOXC2, and VEGFR-3 (*Figure 2—figure supplement 1b,f,g*). Interestingly, while FOXC2 was strongly reduced in collecting vessels of EC-*Foxc2*-KO mice, we also observed a reduction in FOXC1 expression in collecting vessels, and particularly valve regions, compared to littermate controls (*Figure 2—figure supplement 1f and g*). Although chylous effusion was occasionally observed in mutant individuals at P6 (4/30; 13%), inactivation of *Foxc2* beginning at P1 significantly reduced the total number of valves in mesenteric lymphatic collecting vessels (*Figure 2g – l, q, r, w, x*). However, no obvious changes were observed in the mesenteric blood vasculature of *Foxc2* mutants by CD31 immunostaining (*Figure 2g,h*). Quantitative analysis also determined that the percentage of the total lymphatic valves with mature, intraluminal bi-leaflet structures was significantly reduced with inactivation of *Foxc2* and many valve regions were characterized by collecting vessel constriction, indicative of regression of these structures (*Figure 2i – l, x*). Several of these valve regions were characterized by intraluminal leaflets shortened in length compared to littermate controls, yet a majority of valve regions were characterized by near complete degeneration of intraluminal valves (*Figure 2i – l*), similar to our group's observations previously reported in LEC-*Foxc2*-KO mice (*Sabine et al., 2015*). Moreover, expression of α9−integrin was reduced in EC-*Foxc2*-KO mice compared to littermate controls (*Figure 2j – l*), indicating that the phenotype observed in EC-*Foxc2*-KO mice is also associated with impaired valve maturation processes. Thus, our results using a different endothelial-specific *Cre* strain further support the previously reported role for FOXC2 in postnatal lymphatic vascular function and highlight a more prominent role than FOXC1 in valve maintenance.

Our group previously reported overlapping expression patterns of *Foxc1* and *Foxc2* in the embryonic murine cardiovascular system and key, cooperative functional roles during embryonic lymphangiogenesis (*Kume et al., 1998*; *Winnier et al., 1999*; *Kume et al., 2001*; *Seo et al., 2006*; *Fatima et al., 2016*). To assess the requirement and potential cooperative role of both transcription factors in postnatal lymphatic vascular function, we crossed conditional null *Foxc1^fl*; *Foxc2^fl* mice (*Sasman et al., 2012*) with *Cdh5-Cre^ERT2* mice to generate EC-specific *Foxc1*; *Foxc2* mutant (EC-*Foxc1*; *Foxc2*-DKO) mice and administered tamoxifen from P1 to P5 with inactivation in compound mutant individuals verified by qPCR and immunostaining of the mesenteric lymphatic vasculature with antibodies against FOXC1, FOXC2, and VEGFR-3 (*Figure 2—figure supplement 1c,h,i*). At P6, a majority of compound EC-*Foxc1*; *Foxc2* mutant mice (48/78, 61.5%) developed severe chylous ascites (*Figure 2o,p*) and chylothorax (data not shown). PROX1 immunostaining showed that PROX1-high expressing valve regions were nearly absent in compound mutants whereas CD31 immunostaining determined that the blood vasculature appeared normal (*Figure 2m,n,y*). Compared to littermate controls, lymphatic collecting vessel hierarchy of EC-*Foxc1*; *Foxc2*-DKO mutants was markedly abnormal, characterized by high numbers of interconnecting and inter-looping branches resembling a primitive lymphatic vascular plexus prior to valve differentiation, maturation, and remodeling (*Figure 2s,t*). Furthermore, direct comparison of lymphatic collecting vessel hierarchy of EC-*Foxc1*; *Foxc2*-DKO mice with EC-*Foxc2*-KO mice shows that combined deletion of *Foxc1* and *Foxc2* results in a synergistic response (*Figure 2q–t*). Our group previously reported that ectopic sprouting from degenerating valves was observed in LEC-*Foxc2* mutant mice (*Sabine et al., 2015*), thus the markedly abnormal branching pattern observed in EC-*Foxc1*; *Foxc2*-DKO mice is likely a result of accelerated valve regression and ectopic sprouting from these regions to compensate drainage from areas of perturbed flow due to the near complete regression of valves. To verify that *Foxc1* and *Foxc2* inactivation did not simply reduce PROX1 expression in valve regions, we performed additional immunostaining of mesentery tissue for α9-integrin expression. Modest α9-integrin expression was detected in regressing valve leaflets of EC-specific *Foxc2* mutants (*Figure 2—figure supplement 3a,b,e,f*), however expression was absent in collecting vessel branches of EC-*Foxc1*; *Foxc2*-KO mutant where valves are typically present (*Figure 2—figure supplement 3c,d,g,*

*h*). Collectively, we demonstrate that both FOXC1 and FOXC2 are required for postnatal lymphatic vascular function. FOXC2 has a predominant role in maintaining LEC valve identity, however FOXC1 functions in a cooperative manner with FOXC2 to refine valve maturation processes.

### Cell elongation and junctional integrity is markedly impaired in compound EC-specific *Foxc1; Foxc2* mutants and is accompanied by increased apoptosis throughout lymphatic collecting vessels

Collecting vessels of LEC-specific *Foxc2* mutants were previously characterized by the presence of rounded LECs with discontinuous, 'zig-zag' like junctions as opposed to elongated cells with linear, 'zipper-like' junctions aligned with flow. Furthermore, increased apoptosis and inappropriate LEC proliferation was detected in degenerating valve LECs of *Foxc2* mutants, which was associated with abnormal activation of TAZ signaling and impaired regulation of LEC quiescence and survival (*Sabine et al., 2015*). Linearized LEC junctions in collecting vessels help to minimize loss of lymph during its transport (*Baluk et al., 2007*) and disruption of LEC barrier integrity in LEC-*Foxc2*-KO mice contributed to leakage of lymph and the development of chylous ascites and chylothorax (*Sabine et al., 2015*). To first determine how inactivation of both *Foxc1* and *Foxc2* affects collecting vessel LEC junctional integrity, we immunostained mesenteric lymphatic vessels for PROX1 and VE-Cadherin expression (*Figure 2—figure supplement 4a–h* and *Videos 1–4*). As previously reported, LECs in wild-type mice were elongated, organized and aligned with the direction of flow, and had continuous cell-cell junctions (*Figure 2—figure supplement 4a,c,e,g*, *Video 1*, and *Video 3*) whereas EC-specific inactivation of *Foxc2* resulted in a more rounded shape with disrupted endothelial junctions (*Figure 2—figure supplement 4b,f*, and *Video 2*). However, inactivation of both *Foxc1* and *Foxc2* severely impaired LEC organization as cells appeared more spherical in shape and were poorly elongated in alignment with flow (*Figure 2—figure supplement 4d,h*, and *Video 4*). Surprisingly, we not only observed the presence of discontinuous cell-cell junctions in compound *Foxc1; Foxc2* mutant mice, but the distribution of VE-Cadherin at cell junctions was broader compared to littermate controls and *Foxc2* mutants (*Figure 2—figure supplement 4e – h,* and *Video 4*), indicative of an increase in overlapping cell junctions. Because the presence of increased overlapping cell-cell junctions was observed in compound *Foxc1; Foxc2* mutant mice, but not *Foxc2* mutants, this suggests a separate role for FOXC1 and FOXC2 in regulation of LEC intercellular junctions.

Our group previously reported that active caspase 3-positive LECs were detected only in minor instances throughout collecting vessels of control mice, whereas apoptosis in LEC-specific *Foxc2* mutants was more frequently detected and

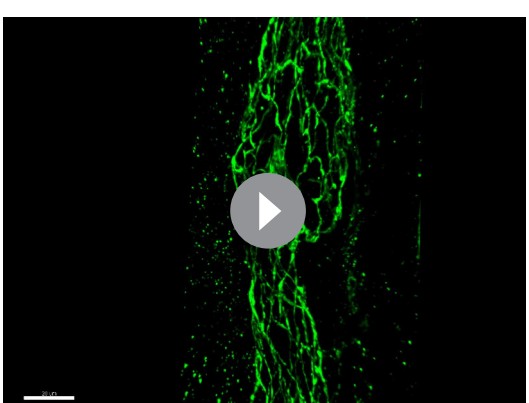

**Video 1.** 3D-visualization of VE-Cadherin expression in a mesentery lymphatic collecting vessel of a *Foxc2fl/fl* mouse. Three-dimensional reconstruction of a P6 *Foxc2fl/fl* control mouse administered tamoxifen from P1-P5, using Imaris 'Surpass' function. Mesenteric collecting vessels were immunostained with antibodies against VE-Cadherin (green).
https://elifesciences.org/articles/53814#video1

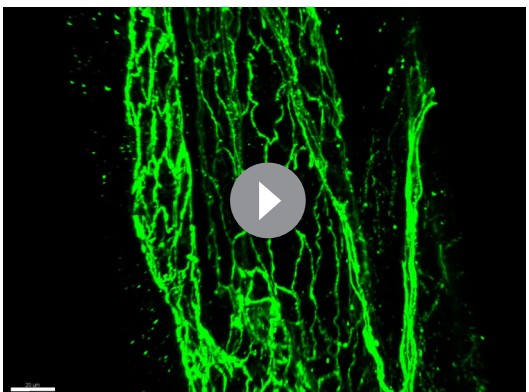

**Video 2.** 3D-visualization of VE-Cadherin expression in a mesentery lymphatic collecting vessel of an EC-*Foxc2*-KO mouse. Three-dimensional reconstruction of a P6 EC-*Foxc2*-KO mouse administered tamoxifen from P1-P5, using Imaris 'Surpass' function. Mesenteric collecting vessels were immunostained with antibodies against VE-Cadherin (green). Blue arrows denote regions of discontinuous cell-cell junctions.
https://elifesciences.org/articles/53814#video2

limited to degenerating valves (*Sabine et al., 2015*). To next investigate the extent and severity of apoptosis in EC-*Foxc1; Foxc2*-DKO mice, we additionally performed immunostaining of mesentery tissue with PROX1 and active caspase-3 antibodies. In contrast to what was previously reported in LEC-*Foxc2*-KO mice, a significantly greater number of apoptotic cell bodies were detected throughout the lymphatic collecting vessels of EC-*Foxc1; Foxc2*-DKO mice compared to littermate controls (*Figure 2—figure supplement 4i–k*). Of note, apoptotic bodies were more frequently observed in closer proximity to branched regions of the lymphatic collecting vessels (*Figure 2—figure supplement 4j*) potentially indicating areas of valve degeneration.

Together, these data indicate that FOXC1 and FOXC2 have a complementary role in regulating lymphatic collecting vessel function as both transcription factors are required to maintain valve and LEC barrier integrity to properly transport lymph and limit leakage.

## FOXC1 expression is induced by unidirectional laminar shear stress, but not by reciprocating shear stress, on the contrary to FOXC2

Embryonic lymphatic valve formation in mouse embryos is initiated at the time of active lymphatic drainage (*Sabine et al., 2012*) and valves are primarily found at sites of vessel branching points and bifurcations where perturbed lymph flow is present (*Kampmeier, 1928*; *Sabine et al., 2012*). *In vitro*, OSS forces acting on LECs upregulate the expression of FOXC2 and GATA2 as well as induce the activation of NFATc1 (*Kazenwadel et al., 2015*; *Sweet et al., 2015*; *Sabine et al., 2012*; *Sabine et al., 2015*). In cultured human arterial ECs, exposure to LSS forces induced expression of both *FOXC1* and *FOXC2* (*Chen et al., 2017*). Furthermore, *in vivo* models of reduced shear stress in the blood vasculature of zebrafish showed reduced expression of *foxc1a* and *foxc1b* (orthologs of mammalian *FOXC1* and *FOXC2*, respectively) compared to controls (*Chen et al., 2017*). To investigate how different shear stress stimuli affect FOXC1 and FOXC2 expression in LECs, we examined cells cultured under static, laminar, or oscillatory flow conditions. As previously observed, OSS strongly increased the expression of FOXC2 and the formation of overlapping cell-cell junctions (*Sabine et al., 2012*; *Sabine et al., 2015*). However, FOXC1 expression was slightly reduced compared to cells cultured under static conditions. In contrast, LSS upregulated both FOXC1 and FOXC2 expression as cells became aligned with unidirectional laminar flow (*Figure 3*). No correlation was observed between FOXC1 and FOXC2 expression under different shear stress conditions (*Figure 3—figure supplement 1a–c*). Therefore, mechanical stress induced by flow differentially regulates LEC FOXC1 and FOXC2 expression (*Figure 3c*). Thus, our *in vitro* data support our *in vivo* observations of differences in FOXC1 and FOXC2 expression pattern observed in valves due to exposure to both disturbed flow in valve sinuses and pulsatile laminar flow on the intraluminal side of leaflets, likely contributing to high expression of both FOXC1 and FOXC2 in cells located at the free edge of valve leaflets (*Figure 1*). Furthermore, the reduction of FOXC1 expression observed in collecting vessels of EC-*Foxc2*-KO mice *in vivo* (*Figure 2—figure supplement 1g*) is likely attributable to reduced flow as a result of perturbed barrier integrity and increased leakiness from lymphatic collecting vessels.

## FOXC1 regulates actin cytoskeletal organization and cell-matrix adhesion

OSS acting on LECs induces the formation of thick cortical actin fibers and perinuclear F-actin stress fiber formation (*Sabine et al., 2012*; *Sabine et al., 2015*). Similarly, increased F-actin expression was detected in lymphatic valve forming cells *in vivo* (*Sabine et al., 2012*). However, *FOXC2* knockdown in cultured LECs induced actomyosin contractility, which was potentiated by OSS leading to impaired cell-cell adhesion (*Sabine et al., 2015*). To investigate potential cytoskeletal signaling changes in the context of loss of *FOXC1* with shear stress, LECs were transfected at subconfluency with scramble control or *FOXC1* siRNAs, then seeded into fibronectin-coated flow chambers under static, LSS, or OSS conditions. *FOXC1* knockdown resulted in impaired adhesion and reduced viability of LECs as the monolayer of cells re-seeded into flow chambers was characterized by several areas devoid of cells and cells were observed floating over the monolayer under static conditions, suggesting cells had been flushed away under LSS and OSS (*Figure 4*). Intriguingly, *FOXC1* inactivation also resulted in a substantial increase of transverse actin stress fiber formation associated with increased phosphorylation of MLC2 (p-MLC2) (*Figure 4a*), indicative of increased actomyosin

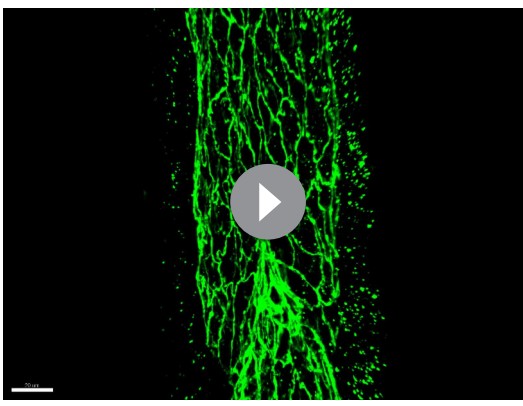

**Video 3.** 3D-visualization of VE-Cadherin expression in a mesentery lymphatic collecting vessel of a *Foxc1^{fl/fl}; Foxc2^{fl/fl}* mouse. Three-dimensional reconstruction of a P6 *Foxc1^{fl/fl}; Foxc2^{fl/fl}* control mouse administered tamoxifen from P1-P5, using Imaris 'Surpass' function. Mesenteric collecting vessels were immunostained with antibodies against VE-Cadherin (green).
https://elifesciences.org/articles/53814#video3

contractility. This was potentiated under both OSS and LSS, with cells also losing their ability to properly orient in the direction of LSS.

Considering that *FOXC1* knockdown in LECs impairs cell adhesion, we slightly modified our protocol: LECs were first seeded on fibronectin-coated surfaces, then transfected with scramble control, *FOXC1*, *FOXC2*, or combined *FOXC1/FOXC2* (50%:50%) siRNAs and kept under static conditions for 2 days. Given that *FOXC2* inactivation and exposure to OSS in LECs results in increased actomyosin contractility (*Sabine et al., 2015*), we investigated whether combined knockdown of *FOXC1* and *FOXC2* in LECs would result in a synergistic effect. *FOXC1*-KD LECs were characterized by strongly increased co-localization of p-MLC2 signal with abnormally accumulated transverse F-actin stress fibers compared to control and *FOXC2*-KD LECs with modest p-MLC2 signal co-localized mostly to the cortical actin ring (*Figure 4b*). Combined knockdown of *FOXC1* and *FOXC2* resulted in a similar phenotype to *FOXC2*-KD LECs (*Figure 4b*), suggesting a rate-limiting effect of FOXC1 in the control of

actomyosin contractility.

As FOXC2 has been shown to be required to maintain continuous cell-cell junctions in the lymphatic endothelium (*Sabine et al., 2015*) and we observed differences in VE-Cadherin distribution between *Foxc2* and compound *Foxc1; Foxc2* mutants *in vivo* (*Figure 2—figure supplement 4*), we investigated possible differences in VE-Cadherin junctions in *FOXC1*-KD and *FOXC2*-KD LECs. Surprisingly, we observed a greater distribution of overlapping VE-Cadherin-positive junctions in *FOXC1*-KD LECs (*Figure 4—figure supplement 1a*). In contrast, as described previously, *FOXC2*-KD LECs were characterized by discontinuous VE-Cadherin junctions (*Figure 4b* and *Figure 4—figure supplement 1a*). However, both overlapping and discontinuous VE-Cadherin-positive junctions were observed in *FOXC1/FOXC2*-KD LECs (*Figure 4—figure supplement 1a*), thus recapitulating our observations *in vivo* (*Figure 2—figure supplement 4d,h*) and indicating separate roles for FOXC1 and FOXC2 regulation of intercellular junctions.

Cell adhesion to ECM is controlled by focal adhesions, which are composed of protein complexes that link the actin cytoskeleton, and especially stress fibers, to the ECM and are critical for translating signals from the ECM environment (*Parsons et al., 2010*). Because *FOXC1* knockdown impaired LEC attachment to fibronectin, we assessed the distribution of two main focal adhesion components, vinculin and paxillin, in *FOXC1*-KD, *FOXC2*-KD, and *FOXC1/FOXC2*-KD LECs to compare the role of FOXC1 and FOXC2 in regulating focal adhesions (*Figure 4c* and *Figure 4—figure supplement 1b*). Knockdown of *FOXC1* mostly induced formation of vinculin-

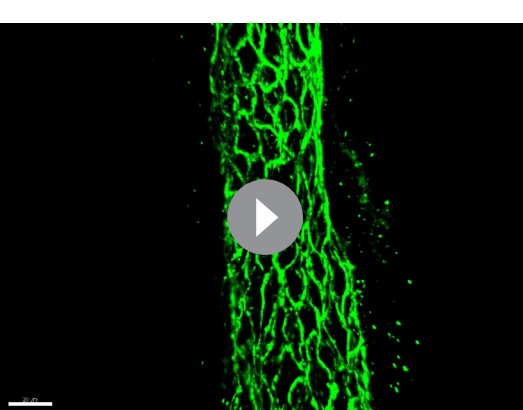

**Video 4.** 3D-visualization of VE-Cadherin expression in a mesentery lymphatic collecting vessel of an EC-*Foxc1; Foxc2*-KO mouse. Three-dimensional reconstruction of a P6 EC-*Foxc1; Foxc2*-KO mouse administered tamoxifen from P1-P5, using Imaris 'Surpass' function. Mesenteric collecting vessels were immunostained with antibodies against VE-Cadherin (green). Blue arrows denote regions of discontinuous cell-cell junctions. Pink arrows denote regions of overlapping cell-cell junctions.
https://elifesciences.org/articles/53814#video4

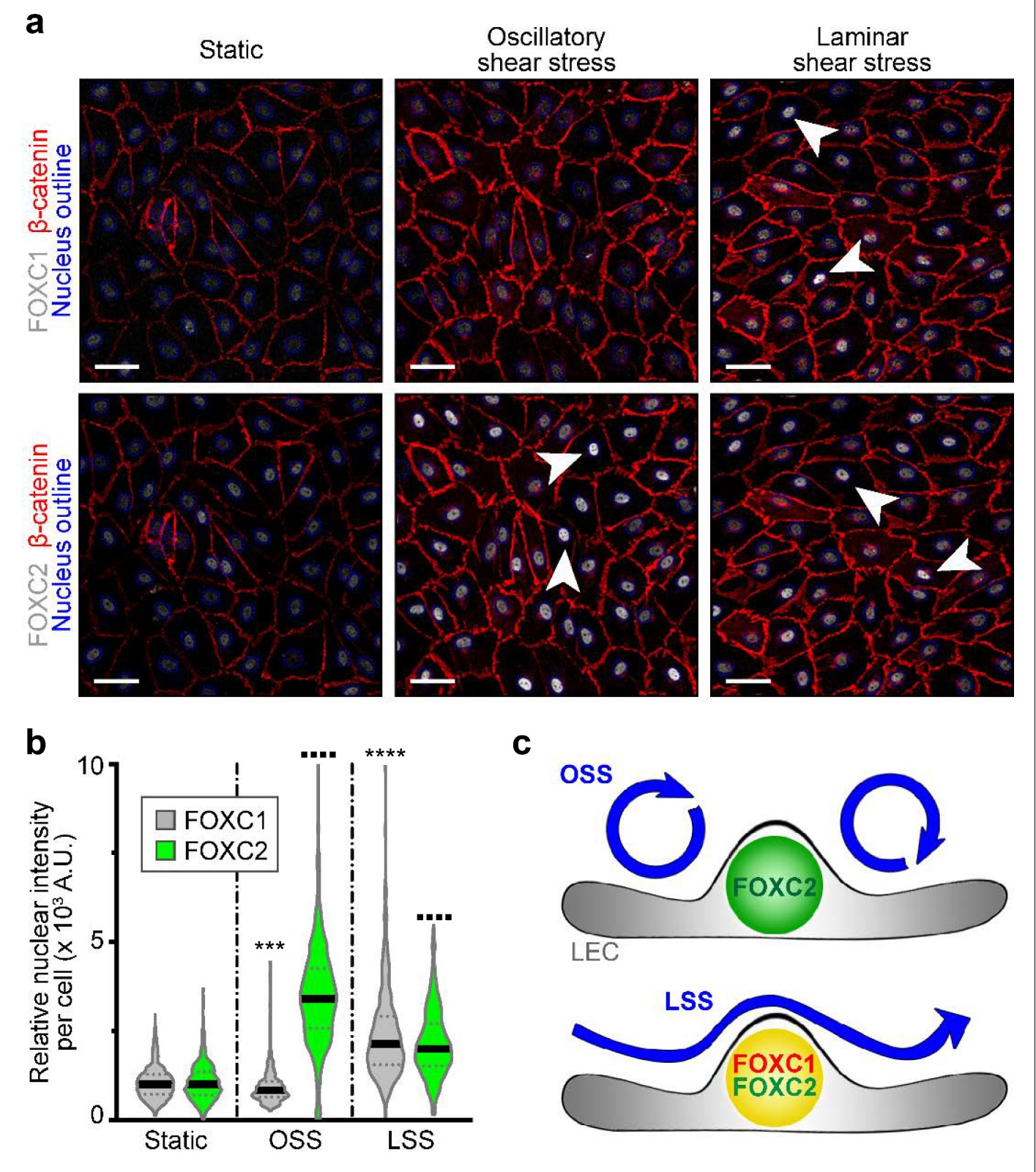

**Figure 3.** FOXC1 expression is induced by unidirectional laminar shear stress, but not by reciprocating shear stress, on the contrary to FOXC2. (a) Representative images of cultured LECs under static, OSS, or LSS show increased expression of FOXC1 when subjected to 24 hr to LSS, whereas FOXC2 is induced by both OSS and LSS. Immunostaining for β-catenin (red), FOXC1 (white, top panels), and FOXC2 (white, bottom panels). Nuclei are outlined with dashed blue lines. Arrowheads denote cells with strong nuclear expression of FOXC1 or FOXC2. Scale bars are 10 μm. (b) Corresponding

*Figure 3 continued on next page*

*Figure 3 continued*

quantification of FOXC1 or FOXC2 nuclear intensity per cell (100–200 cells were quantified per condition). Data are presented as violin plots with median values indicated by solid lines and are representative of 3 independent experiments. P-values were obtained using mixed-effects analysis. ***p<0.001, ****p<0.0001 to Static FOXC1 and ■■■■ p<0.0001 to Static FOXC2 . (**c**) Scheme summarizing the observed regulation of FOXC1 and FOXC2 by flow-mediated shear stress: cells under OSS have high levels of FOXC2, whereas cells under LSS have high levels of both FOXC1 and FOXC2.

The online version of this article includes the following figure supplement(s) for figure 3:

**Figure supplement 1.** Absence of correlation of FOXC1 and FOXC2 expression in relation to mechanical stress, and validation of FOXC1 and FOXC2 antibodies and siRNA used in this study.

positive or paxillin-positive focal adhesions at the tip of actin stress fibers (*Figure 4c* and *Figure 4—figure supplement 1b*, respectively), whereas knockdown of *FOXC2* mostly induced formation of vinculin-positive or paxillin-positive adherens junctions associated with the cortical actin ring (*Figure 4c* and *Figure 4—figure supplement 1b*, respectively). Interestingly, combined knockdown of *FOXC1* and *FOXC2* induced formation of both focal adhesions and adherens junctions in LECs (*Figure 4c*, *Figure 4—figure supplement 1b*). Collectively, our data demonstrate that FOXC1 and FOXC2 have separate, yet complementary, roles to regulate LEC cytoskeletal organization and contractility as FOXC1 controls focal adhesions and FOXC2 regulates adherens junctions.

## FOXC1 and FOXC2 regulate LEC expression of negative RhoA signaling regulators PRICKLE1, ARHGAP21, and ARHGAP23

The RhoA-Rho-associated protein kinase (ROCK)-phosphorylated myosin light chain (pMLC) signaling pathway has been well established as a regulator of cytoskeletal contractility mechanisms (*Ridley, 2001*). Furthermore, perturbed RhoA/ROCK signaling activation has been demonstrated to be a negative regulator of blood endothelial tubulogenesis (*Bowers et al., 2016*) and barrier function (*Spindler et al., 2010*). Given the synergistic effect of combined inactivation of *Foxc1* and *Foxc2* on lymphatic valve maintenance and maturation *in vivo* and perturbed cytoskeletal organization upon *FOXC1* and *FOXC2* knockdown *in vitro*, we hypothesized that FOXC1 and FOXC2 transcription targets may converge to negatively regulate RhoA/ROCK activation in the context of mechanical stress. To identify potential negative regulators of RhoA/ROCK downstream of FOXC1 and FOXC2 transcription, we utilized RNA-seq data previously published from our LEC- *Foxc1; Foxc2*-DKO mice (*Fatima et al., 2016*). Recent evidence has shown that a signaling complex, consisting of the planar cell polarity (PCP) protein Prickle1 and the Rho GTPase activating proteins (GAPs) Arhgap21 and Arhgap23, functions to inhibit RhoA on the non-protrusive lateral membrane cortex allowing for dynamic cell morphology necessary for proper migration (*Zhang et al., 2016*). Our RNA-seq analysis (*Fatima et al., 2016*) revealed that mRNA expression of *Prickle1*, *Arhgap21*, and *Arhgap23* was significantly downregulated in dermal LECs isolated from LEC- *Foxc1; Foxc2*-DKO mouse embryos (*Figure 5a*). Furthermore, our RNA-seq analysis showed a modest, yet significant reduction of RhoA, but other GAPs associated with regulation of RhoA, endothelial barrier function, and lumen maintenance (*van Buul et al., 2014*; *Barry et al., 2016*) were not differentially expressed with the exception of a significant increase in *Arhgap18* expression and a modest reduction of *Arhgap20* expression (*Figure 5—figure supplement 1k*). Because of the previous report demonstrating a physical interaction of Prickle1 and Arhgap21/23 complex (*Zhang et al., 2016*) and our observation that expression of all three genes was significantly downregulated in our RNA-seq analysis, we focused our attention on these potential downstream targets. Therefore, we performed in silico analysis to identify putative FOXC binding sites in the human *PRICKLE1*, *ARHGAP21*, and *ARHGAP23* loci. First, the Hypergeometric Optimization of Motif EnRichment (HOMER) suite of tools (*Heinz et al., 2010*) was used to identify regions containing the conserved RYMAAYA FOX consensus binding sequence (*Pierrou et al., 1994*; *De Val et al., 2008*; *Norrmén et al., 2009*) in active areas of transcription identified with the Encyclopedia of DNA Elements (ENCODE) (2012) and the UCSC Human Genome Browser (*Kent et al., 2002*). Then, we utilized the Evolutionary Conserved Region (ECR) Browser (https://ecrbrowser.dcode.org) (*Ovcharenko et al., 2004*) tool to identify conserved binding regions between the mouse and human genomes. Several conserved and aligned putative FOX binding regions were identified in regions of the *PRICKLE1* (*Figure 5b,c*), *ARHGAP21*,

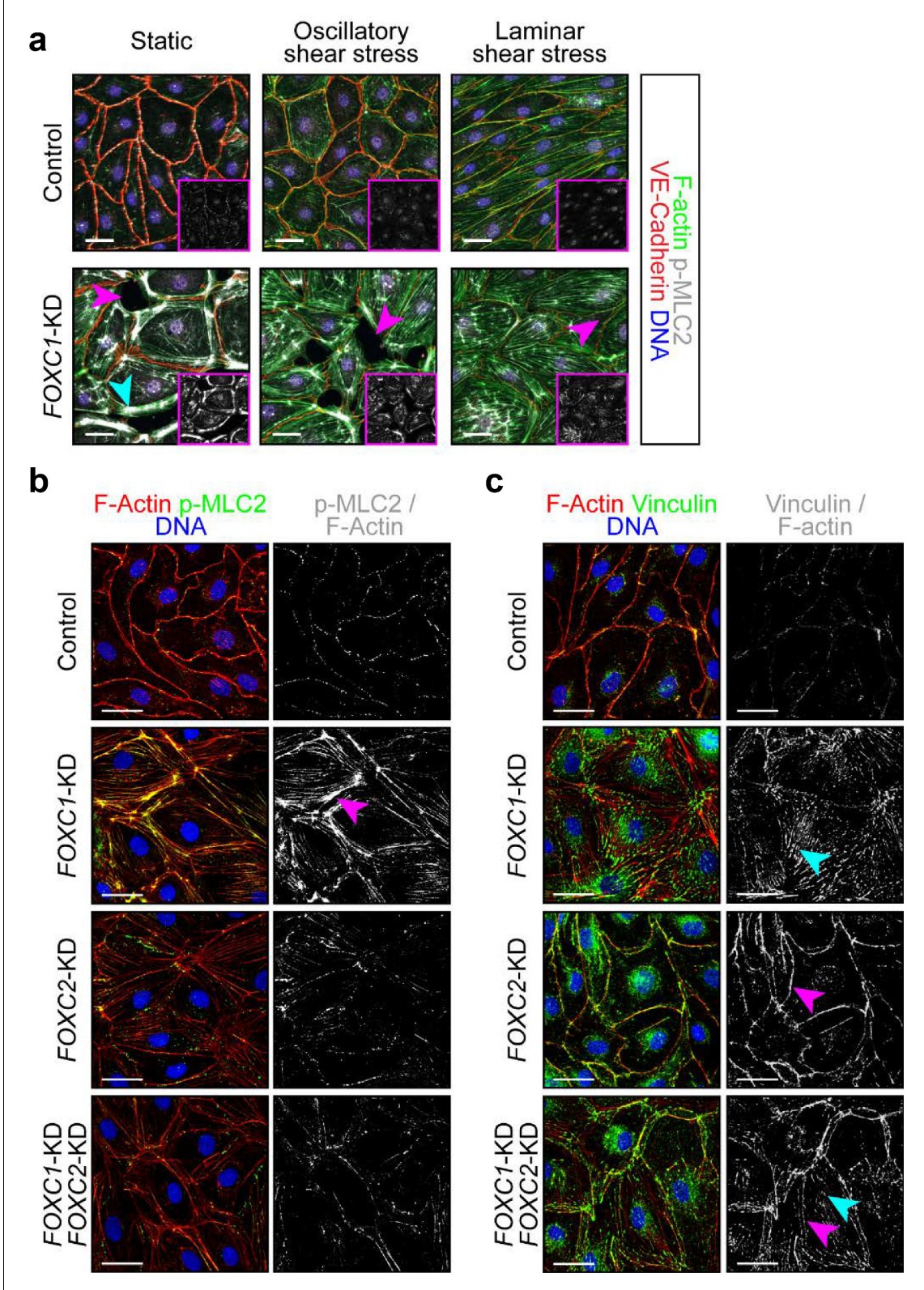

**Figure 4.** FOXC1 regulates actin cytoskeletal organization and cell-matrix adhesion. (**a**) Representative images of Control and FOXC1-KD LECs cultured under static, OSS, or LSS show reduced viability and higher number of contractile stress fibers (cyan arrowhead). Immunostaining for VE-Cadherin (red), F-actin (green), p-MLC2 (white), and DNA (blue). Pink arrowheads denote areas devoid of cells in the endothelial monolayer. Pink inserts denote single-channel p-MLC2 (white) images. Scale bars are 30 μm. (**b, c**) Representative images of Control, *FOXC1*-KD, *FOXC2*-KD, and combined *FOXC1*-KD/

*Figure 4 continued on next page*

*Figure 4 continued*

*FOXC2*-KD LECs show higher number of contractile stress fibers (b, pink arrowhead) and focal adhesions (c, cyan arrowhead) upon *FOXC1* knockdown. In comparison, *FOXC2* knockdown rather induced focal adherens junctions (c, pink arrowhead). (b) Immunostaining for F-actin (red), p-MLC2 (green), and DNA (blue). Images on the right show a mask applied to visualize only double F-actin$^+$/p-MLC2$^+$ (white) stress fibers. Scale bars are 30 μm. (c) Immunostaining for F-actin (red), vinculin (green), and DNA (blue). Images on the right show a mask applied to visualize only double F-actin$^+$/vinculin$^+$ (white) adhesion sites. Scale bars are 30 μm.

The online version of this article includes the following figure supplement(s) for figure 4:

**Figure supplement 1.** FOXC1 regulates intercellular junctions, actin cytoskeletal organization and cell-matrix adhesion.

and *ARHGAP23* (*Figure 5—figure supplement 1a,b*) loci. To identify direct interaction of FOXC1 and FOXC2 with these putative binding regions, human dermal LECs from juvenile foreskin (HDLECs) were cultured, and ChIP assays were performed using two specific antibodies for FOXC1 and one antibody for FOXC2 for regions identified in the *PRICKLE1* (*Figure 5d*), *ARHGAP21*, and *ARHGAP23* loci (*Figure 5—figure supplement 1c,d*). We found that binding of both FOXC1 and FOXC2 were significantly enriched in ECR-2, and −6 in *PRICKLE1* whereas specific binding of FOXC1 was significantly enriched in ECR-5 and FOXC2 in ECR-1 (*Figure 5e–j*). Additionally, FOXC1 and FOXC2 binding was significantly enriched in ECR-1–1 of the *ARHGAP21* locus and FOXC2 was significantly enriched in ECR-1–2 (*Figure 5—figure supplement 1e–g*). Finally, only FOXC2 binding was significantly enriched in ECR-1 of the *ARHGAP23* locus (*Figure 5—figure supplement 1h*).

Although both FOXC1 antibodies recognize the same immunogen peptide, we observed notable differences in ChIP signals between the two antibodies. However, significant variations dependent on the antibody manufacturer have been previously noted for antibodies targeted to the same antigen in immunohistochemistry related work (*Ramos-Vara, 2005*). To validate our ChIP assays, we also performed a negative control experiment using primers to amplify a portion of the promoter region of *ICAM1* that was not predicted to bind FOX transcription factors by in silico analysis (*Figure 5—figure supplement 1i*). ChIP analysis using the antibodies against FOXC1, FOXC2, and IgG for the *ICAM1* region from three experimental replicates showed no band signals (*Figure 5—figure supplement 1j*) validating specificity of FOXC1 and FOXC2 binding to these loci in LECs. Thus, these results demonstrate that FOXC1 and FOXC2 have similar binding capacities for the *PRICKLE1* and *ARHGAP21* loci whereas FOXC2 shows specificity for the *ARHGAP23* locus.

To assess whether Prickle1 is regulated by FOXC1 and FOXC2 in lymphatic valves *in vivo*, we characterized its expression in lymphatic valves during postnatal development. Prickle1 expression was concentrated on the free edges of PROX1-high valve leaflets located at or near the valve buttress (*Sabine et al., 2018*) with modest expression detected in other leaflet cells (*Figure 5k,m,o*, *Figure 5—figure supplement 2a–d,g,h*, *Video 5*, and *Video 6*). Postnatal deletion of *Foxc1* reduced Prickle1 expression within leaflet free-edge cells while other valve leaflet LECs retained modest expression (*Figure 5l*, *Figure 5—figure supplement 2e,f,i* and *Video 7*). In contrast, postnatal deletion of *Foxc2* or both *Foxc1* and *Foxc2* resulted in a strong and broad reduction of Prickle1 within the lymphatic vasculature compared to littermate controls (*Figure 5m–p*). Collectively, these data show that *PRICKLE1* is a novel target of FOXC transcription factors in valve forming LECs.

## ROCK inhibition rescues hypercontractility of actin cytoskeleton in *FOXC1*-KD and *FOXC2*-KD LECs

To investigate whether ROCK signaling is activated in the absence of FOXC1 and FOXC2, LECs were transfected with scramble Control, *FOXC1*, or *FOXC2* siRNAs under static conditions and treated with vehicle Control or the ROCK inhibitor Y-27632 (*Figure 6*). Immunostaining and quantification of F-actin and p-MLC2 demonstrated an increase in the F-actin and p-MLC2 area per cell with loss of *FOXC1* and treatment with vehicle. However, inhibition of ROCK reduced F-actin and pMLC2 distribution to a similar level in Control LECs (*Figure 6a,b*). In contrast, loss of *FOXC2* modestly increased F-actin and pMLC2 area per cell, but the increase in relative area per cell was greatly potentiated by OSS. However, similar to *FOXC1* knockdown, changes in F-actin and p-MLC2 area were ameliorated by treatment with Y-27632 (*Figure 6c,d*). Together, this data demonstrate that key targets downstream of FOXC1 and FOXC2 transcription activity function in part to negatively regulate RhoA/ROCK activation in LECs.

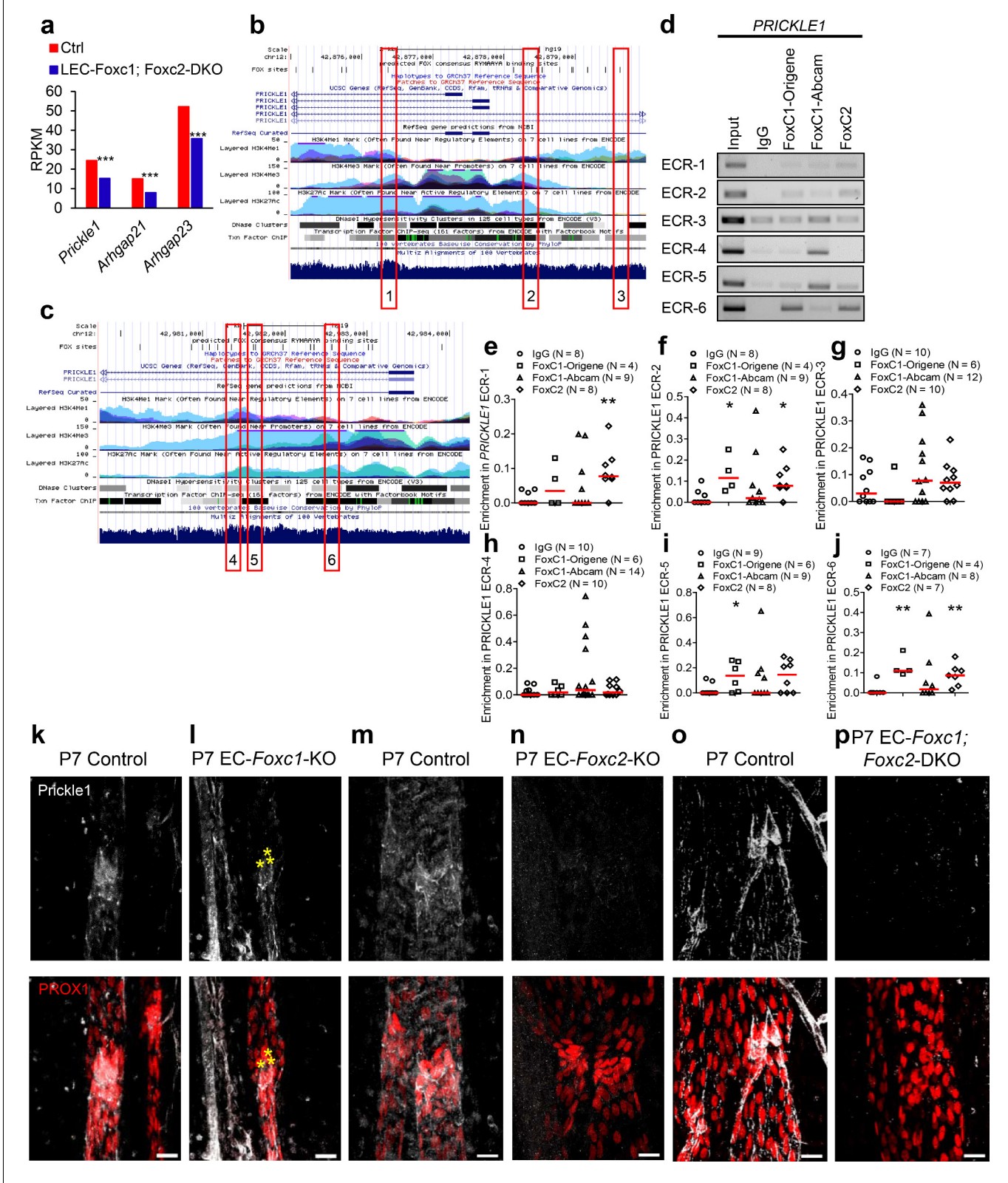

**Figure 5.** FOXC1 and FOXC2 mediate lymphatic expression of the PCP signaling component and RHOA signaling regulator Prickle1. (a) Reduced expression of *Prickle1*, *Arhgap21*, and *Arhgap23* in *Foxc1/c2*-compound mutant LECs isolated from the dorsal skin at E15.5. Graph shows RPKM values from RNA-seq analysis. *** denotes p<0.001. (b, c) Putative FOX-binding sites in regions of the human *PRICKLE1* locus as viewed on the UCSC genome browser (http://genome.ucsc.edu; ***Kent et al., 2002***). Vertical lines on the 'FOX sites' track indicate putative FOX binding sites predicted using HOMER

*Figure 5 continued on next page*

*Figure 5 continued*

(see methods). Red boxes indicate evolutionary conserved regions (ECRs) containing FOX-binding sites between human and mouse genomes that are conserved and aligned. (d) ChIP showing specific binding of FOXC1 and FOXC2 to the consensus FOX-binding sites within ECRs 1, 2, 3, 4, 5, and six in *PRICKLE1* in HDLECs. (e–j) ChIP assays were performed using antibodies against Foxc1, Foxc2, and normal goat IgG. The binding of FOXCs to candidate ECRs identified in b and c in the *PRICKLE1* locus was determined with regular PCR and expressed as relative folds of input whose band intensity was normalized to 1. Data are presented as a scatter plot with median indicated in red. * denotes p<0.05, ** denotes p<0.01 as determined by Mann-Whitney two-tailed test. (k–p) Representative images of lymphatic valve regions in mesenteric collecting vessels immunostained with antibodies targeted to Prickle1 and PROX1 in P7 *Foxc1* Control (k), EC-*Foxc1*-KO (l), *Foxc2* Control (m), EC-*Foxc2*-KO, (n) *Foxc1; Foxc2* control and (o) EC-*Foxc1; Foxc2*-DKO (p) mice show reduced Prickle1 expression in the valve leaflets and lymphangion of *Foxc2* and compound *Foxc1; Foxc2* mutants compared to littermate controls whereas Prickle1 is reduced primarily in leaflet-free-edge LECs, denoted by yellow asterisks, of *Foxc1* mutants. Scale bars are 20 μm.

The online version of this article includes the following figure supplement(s) for figure 5:

**Figure supplement 1.** FOXC1 and FOXC2 mediate lymphatic expression of the RHO GTPase activating proteins Arhgap21 and Arhgap23.
**Figure supplement 2.** Prickle1 is more highly expressed in the leaflet-free-edge LECs of lymphatic valves.

## Inhibition of ROCK is able to partially rescue lymphatic valve degeneration in EC-specific *Foxc2* mutant mice but not compound *Foxc1; Foxc2* mutant mice

As ROCK inhibition rescued impaired cytoskeletal organization in *FOXC1*- and *FOXC2*-KD LECs, we assessed whether ROCK inhibition could rescue lymphatic valve degeneration observed in EC-*Foxc2*-KO and EC-*Foxc1; Foxc2*-DKO mice as LEC-specific deletion of *Foxc1* and *Foxc2* did not affect *Rock1* and *Rock2* expression (*Figure 5—figure supplement 1k*). Compared to P7 *Foxc2* mutants administered DPBS, *Foxc2* mutants administered Y-27632 concurrently with tamoxifen from P2 – P5 were able to retain a significantly higher total valve number, similar to totals quantified in littermate control mice injected with DPBS or Y-27632 (*Figure 7a–d,i*). However, the proportion of mature lymphatic valves present in *Foxc2* mutants after ROCK inhibition was significantly lower compared to littermate controls treated with DPBS or Y-27632, but significantly increased compared to *Foxc2* mutants administered DPBS (*Figure 7j*). In contrast, ROCK inhibition was not able to rescue near complete lymphatic valve degeneration in P7 EC-*Foxc1; Foxc2*-DKO mutants (*Figure 7e–h, k*).

## Inhibition of ROCK improves LEC junctional integrity in collecting vessels of EC-*Foxc2*-KO and EC-*Foxc1; Foxc2*-DKO mice

Although ROCK inhibition was not able to completely rescue the valve phenotypes associated with EC-*Foxc2*-KO nor *EC-Foxc1; Foxc2*-DKO mice, we assessed the effect of ROCK inhibition on LEC junctional integrity in collecting vessels of both mutant mouse lines as Y-27632 treatment rescued impaired LEC cytoskeletal organization *in vitro*. Immunostaining of VE-Cadherin in collecting vessels of P7 EC-*Foxc2*-KO and EC-*Foxc1; Foxc2*-DKO mutant mice treated with DPBS identified disrupted LEC cell-cell junctions (*Figure 8a,b,e,f*) similar to our observations in untreated mutant mice at P6 (*Figure 2—figure supplement 4*) contributing to loss of lymphatic vascular barrier function. However, ROCK inhibition in both EC-*Foxc2*-KO and EC-*Foxc1; Foxc2*-DKO mutant mice was able to in part restore continuous cell-cell 'zipper-like' junctions characteristic of lymphatic collecting

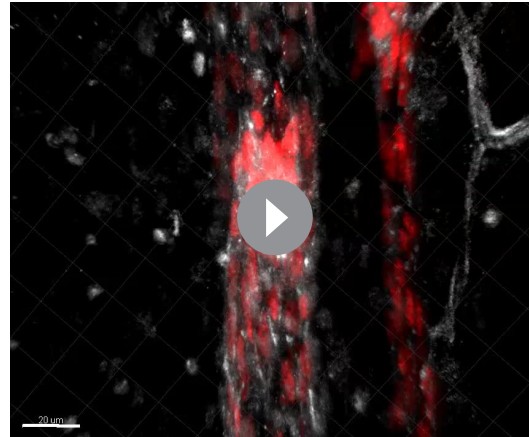

**Video 5.** 3D-visualization of Prickle1 expression in lymphatic valve leaflet free-edges in a mesentery collecting vessel of a *Foxc1*<sup>fl/fl</sup> mouse. Three-dimensional reconstruction of a P7 *Foxc1*<sup>fl/fl</sup> control mouse administered tamoxifen from P1-P5, using Imaris 'Surpass' function. Mesenteric collecting vessels were immunostained with antibodies against Prox1 (red) and Prickle1 (white). Note Pricke1 is more highly expressed within LECs at the free-edges of valve leaflets and at the valve buttress.
https://elifesciences.org/articles/53814#video5

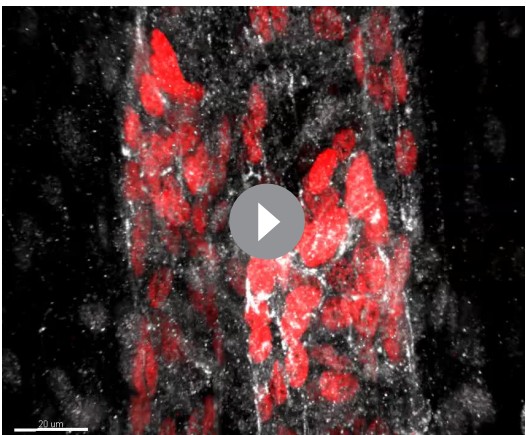

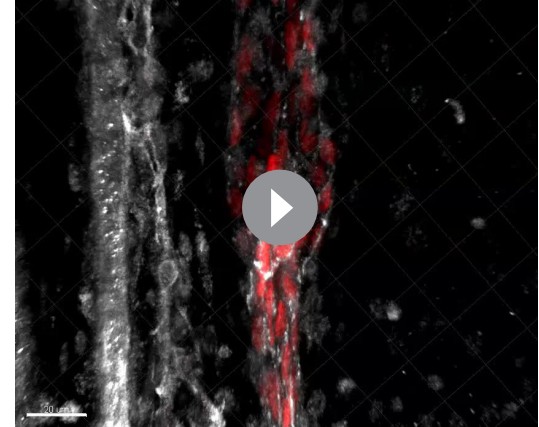

**Video 6.** 3D-visualization of Prickle1 expression in lymphatic valve leaflet free-edges in a mesentery collecting vessel of a *Foxc2*^fl/fl mouse. Three-dimensional reconstruction of a P7 *Foxc2*^fl/fl control mouse administered tamoxifen from P1-P5, using Imaris 'Surpass' function. Mesenteric collecting vessels were immunostained with antibodies against Prox1 (red) and Prickle1 (white). Note Pricke1 is more highly expressed within LECs at the free-edges of valve leaflets and at the valve buttress of both Prox1-hi valve regions.

https://elifesciences.org/articles/53814#video6

**Video 7.** 3D-visualization of reduced Prickle1 expression in lymphatic valve leaflet free-edges in a mesentery collecting vessel of an EC-*Foxc1*-KO mouse. Three-dimensional reconstruction of a P7 EC-Foxc1-KO mouse administered tamoxifen from P1-P5, using Imaris 'Surpass' function. Mesenteric collecting vessels were immunostained with antibodies against Prox1 (red) and Prickle1 (white). Note the reduction of Prickle1 expression within LECs at the leaflet free-edge compared to *Videos 1* and *2*, whereas Prickle1 expression is retained in the valve sinus regions.

https://elifesciences.org/articles/53814#video7

vessels (*Figure 8c,d,g,h*), thus resulting in improved junction integrity. Collectively, our *in vitro* and *in vivo* data demonstrate that FOXC1 and FOXC2 function in part to negatively regulate perturbed activation of Rho/ROCK signaling in LECs downstream of shear stress-sensing mechanisms.

## FOXC2 can functionally substitute for FOXC1 during lymphatic valve development, maintenance, and maturation

Because the forkhead DNA binding domain is nearly identical between FOXC1 and FOXC2 and both transcription factors have cooperative roles in early cardiovascular development (*Kume, 2009*; *Lam et al., 2013*), we assessed whether FOXC2 could substitute for FOXC1 transcriptional activity *in vivo* in the context of lymphatic valve development, maintenance, and maturation. To do so, we generated mice that carry a *Foxc2* knock-in (*Foxc1*^c2) allele, in which the *Foxc1* coding region has been replaced with the cDNA coding (from the start codon to the stop codon) for *Foxc2* (*Figure 9a – c*). By breeding male and female *Foxc1*^c2/+ mice, we were able to generate litters consisting of *Foxc1*^+/+, *Foxc1*^c2/+, and *Foxc1*^c2/c2 mice for analysis. To verify the absence of FOXC1 expression within the lymphatic vessels of this model, we first performed co-immunostaining of mesentery tissue from P12 *Foxc1*^+/+ and littermate *Foxc1*^c2/c2 mice for FOXC1 and VEGFR-3 (*Figure 9d and e*), which confirmed the complete reduction of FOXC1 expression in lymphatic collecting vessels in homozygous *Foxc2* knock-in mutant mice (*Figure 9e*). To further characterize changes in expression of FOXC1 and FOXC2 in both the blood and lymphatic vasculature, we also performed co-immunostaining for FOXC1, FOXC2, and VE-Cadherin in mesentery tissue of littermate P6 *Foxc1*^+/+ controls and *Foxc1*^c2/+ and *Foxc1*^c2/c2 mutant mice (*Figure 9f – k*). Within lymphatic valves of mesentery collecting vessels, nuclear FOXC1 expression was strongly reduced in *Foxc1*^c2/+ mutants compared to *Foxc1*^+/+ controls. Furthermore, *Foxc1*^c2/c2 mutants were absent of nuclear FOXC1 expression within LECs, although a non-specific, positive signal was detected in the membrane of valve leaflets of both *Foxc1*^c2/+ and *Foxc1*^c2/c2 mutant mice (*Figure 9f – h*). Additionally, co-immunostaining of FOXC1, FOXC2, and VE-Cadherin demonstrated loss of FOXC1 expression in the blood vasculature and smooth muscle cells associated with mesenteric arteries and subsequent gain of FOXC2 expression in *Foxc1*^c2/+ and *Foxc1*^c2/c2 mutants with no obvious changes observed in the blood vasculature

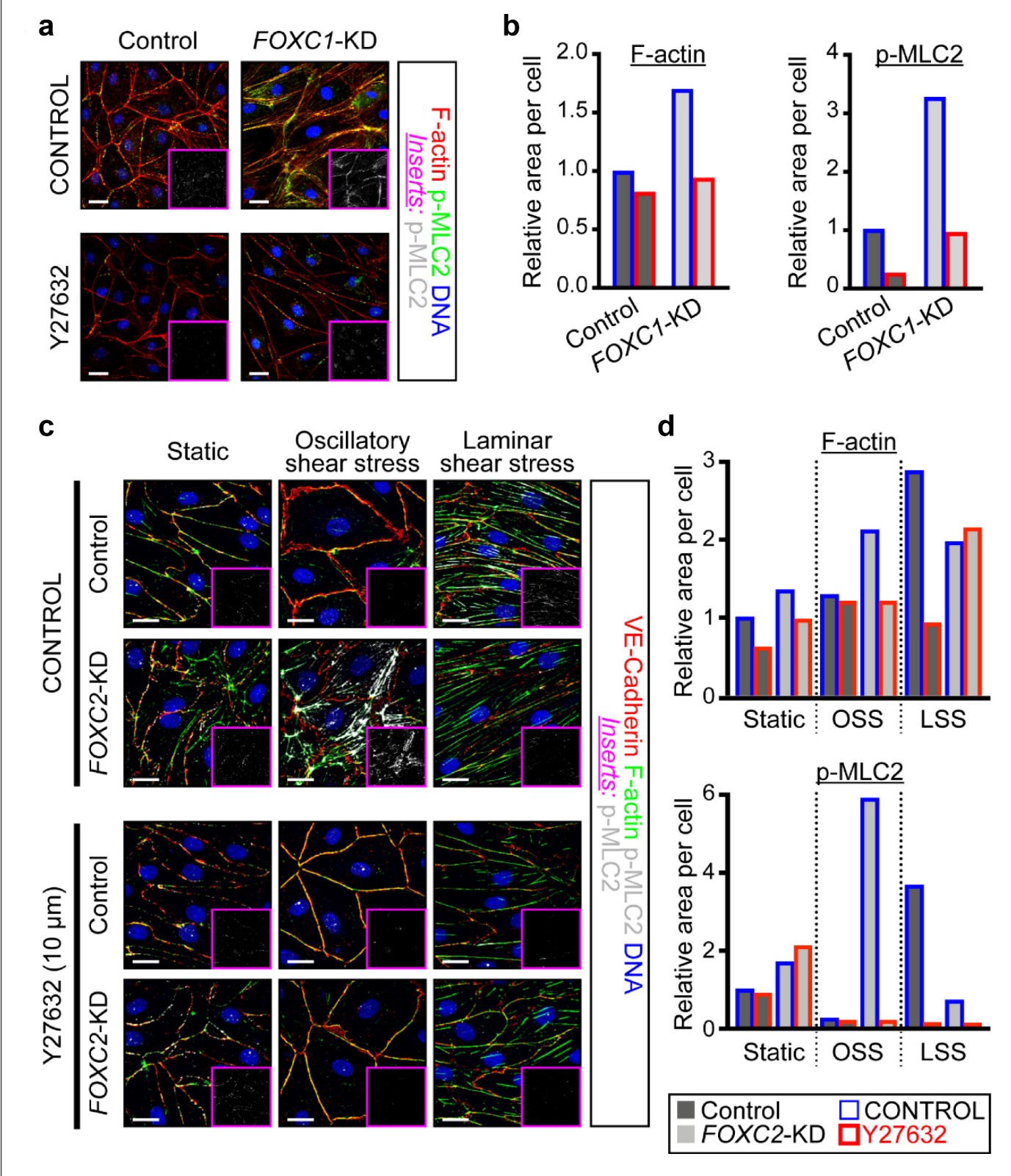

**Figure 6.** ROCK inhibition rescues hypercontractility of actin cytoskeleton in *FOXC1*-KD and *FOXC2*-KD LECs. (a) Representative images of cultured LECs transfected with scramble control siRNA or *FOXC1* siRNA and treated with vehicle control or Y-27632 (10 μM). Treatment with Y-27632 ROCK inhibitor shows rescue of cytoskeletal changes induced by FOXC1 inactivation. Immunostaining of F-actin (red), p-MLC2 (green), and DNA (blue). Pink inserts denote the single-channel p-MLC2 (white) images. Scale bars are 20 μm. (b) Quantification of relative F-actin (left) and p-MLC2 (right) area per

*Figure 6 continued on next page*

*Figure 6 continued*

Control (dark grey bars) and *FOXC1*-KD (light grey bars) cells treated with vehicle Control (blue outline) or Y-27632 (red outline). (c) Representative images of Control and FOXC2-KD LECs cultured under static, OSS, or LSS, treated with vehicle Control or Y-27632 (10 µM). Treatment with Y-27632 ROCK inhibitor shows rescue of cytoskeletal changes induced by FOXC2 inactivation that are most prominent under OSS. Immunostaining for VE-Cadherin (red), F-actin (green), p-MLC2 (white), and DNA (blue). Pink inserts denote the single-channel p-MLC2 (white) images. Scale bars are 20 µm. (d) Quantification of relative F-actin (top) and p-MLC2 (bottom) area per Control (dark grey bars) and *FOXC1*-KD (light grey bars) cells treated with vehicle Control (blue outline) or Y-27632 (red outline).

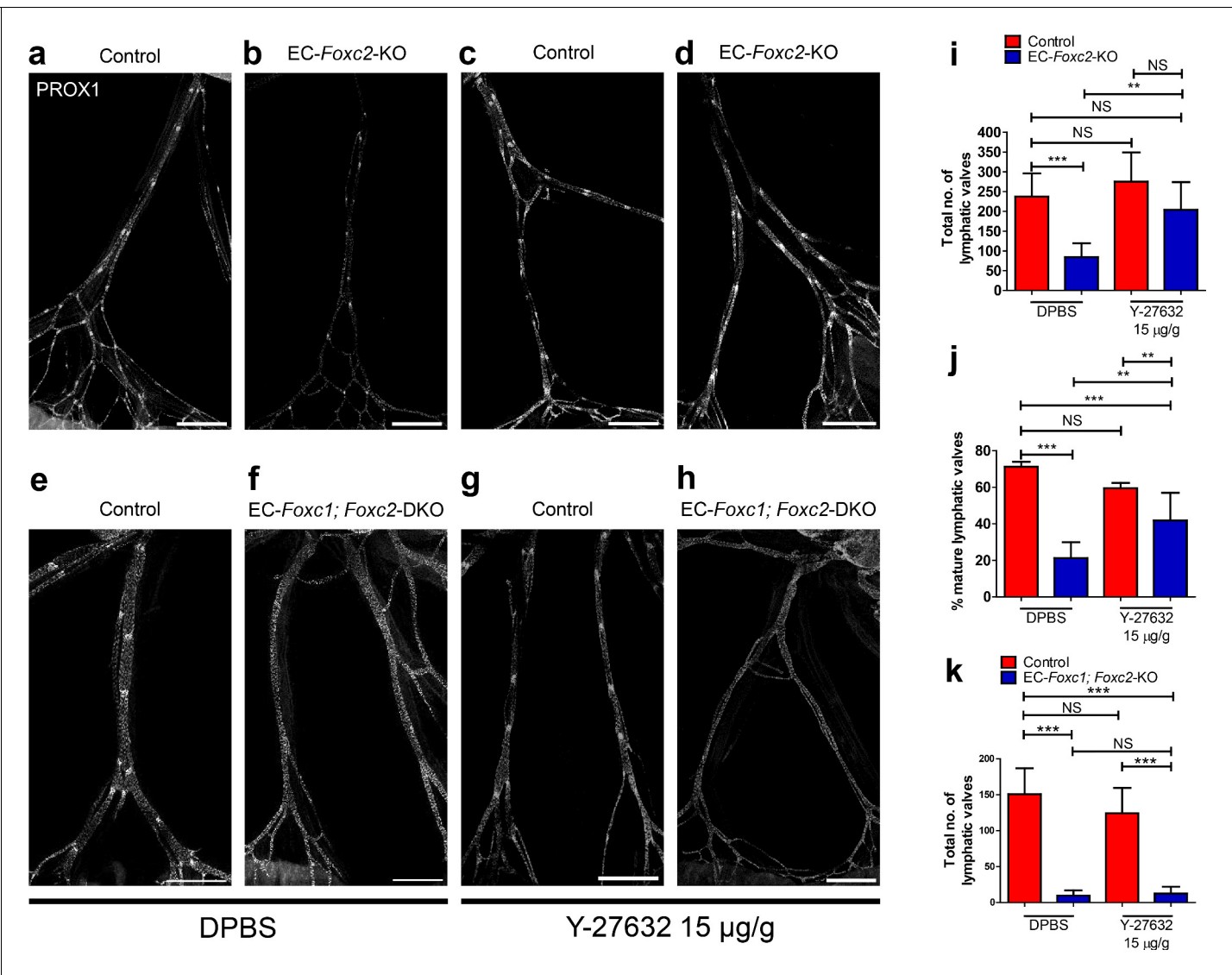

**Figure 7.** Inhibition of ROCK partially rescues postnatally induced valve degeneration in EC-specific *Foxc2* mutant mice but not compound *Foxc1; Foxc2* mutant mice. (a–h) Representative images of lymphatic collecting vessels immunostained with PROX1 antibody in P7 littermate Control (a) and EC-*Foxc2*-KO (b), littermate Control (e) and EC-*Foxc1; Foxc2*-DKO (f) mice subcutaneously injected with DPBS vehicle or littermate Control (c) and EC-*Foxc2*-KO (d), littermate Control (g) and EC-*Foxc1; Foxc2*-DKO (h) mice subcutaneously injected with ROCK inhibitor Y-27632. Scale bars equal to 500 µm. (i, j) Quantification of total valve number (i) and percentage of mature valves (j) in littermate Control and EC-*Foxc2*-KO mice administered DPBS or Y-27632. N = 6 for Control DPBS, N = 6 for EC-*Foxc2*-KO DPBS, N = 9 for Control Y-27632, and N = 9 for EC-*Foxc2*-KO Y-27632. (k) Quantification of total valve number in littermate Control and EC-*Foxc1; Foxc2*-DKO mice administered DPBS or Y-27632. N = 5 for Control DPBS, N = 7 for EC-*Foxc1; Foxc2*-DKO DPBS, N = 8 for Control Y-27632, and N = 9 for EC-*Foxc1; Foxc2*-DKO Y-27632. P-values were obtained by One-way ANOVA with Tukey's post test. Data are presented as mean (± SD). ** denotes p<0.01. *** denotes p<0.001. NS denotes no significance.

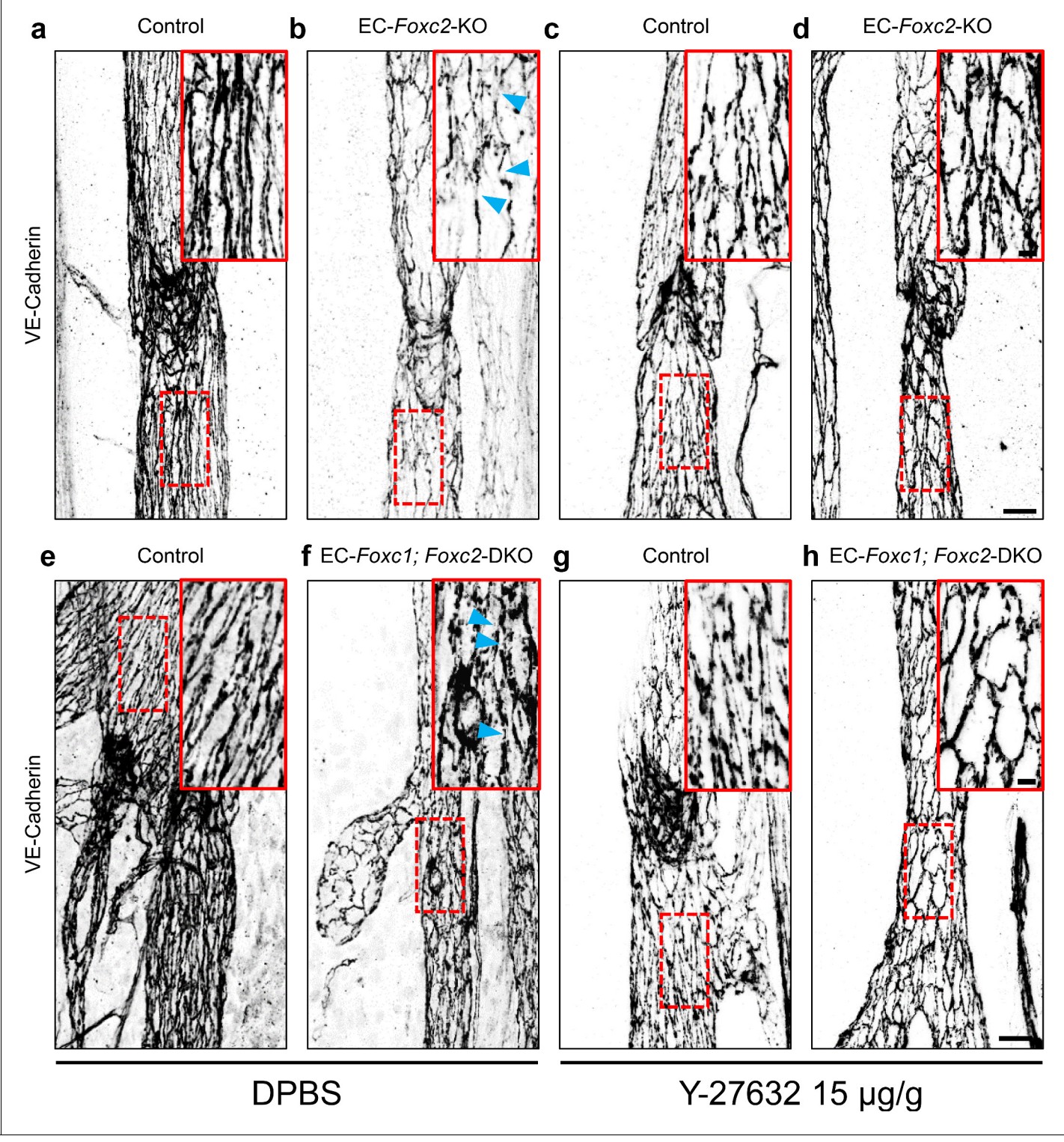

**Figure 8.** Inhibition of ROCK rescues discontinuous cell-cell junctions in EC-specific *Foxc2* and compound *Foxc1; Foxc2* mutant mice. (a–h) High-magnification, representative images of lymphatic collecting vessels immunostained with VE-Cadherin antibody in P7 littermate Control (a) and EC-*Foxc2*-KO (b), littermate Control (e) and EC-*Foxc1; Foxc2*-DKO (f) mice subcutaneously injected with DPBS vehicle or littermate Control (c) and EC-*Foxc2*-KO (d), littermate Control (g) and EC-*Foxc1; Foxc2*-DKO (h) mice subcutaneously injected with ROCK inhibitor Y-27632. Scale bars are 20 μm. Red, dashed boxes highlight magnified regions shown inset. Scale bars equal to 5 μm. Gaps in LEC junctions are visible in EC-*Foxc2*-KO and EC-*Foxc1; Foxc2*-DKO mice administered DPBS (b, f) whereas inhibition of ROCK with Y-27632 is able to restore linear junctions in part (d, h). Blue arrowheads denote discontinuous LEC VE-Cadherin junctions.

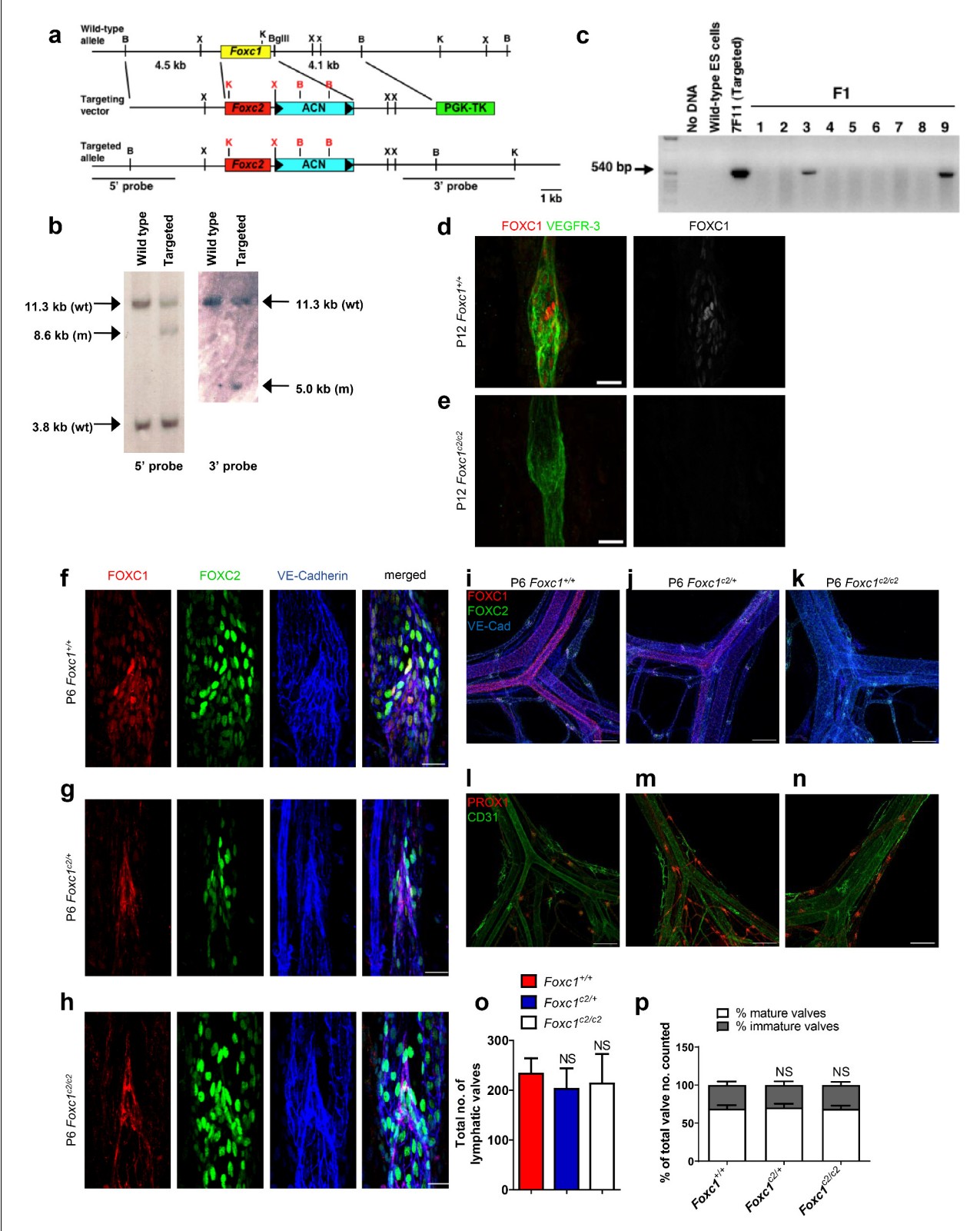

**Figure 9.** FOXC2 is able to functionally substitute for FOXC1 during lymphatic valve development. (a) Schematic representation of the targeting vector and targeted allele. The entire protein coding region of *Foxc1* is replaced with that of *Foxc2*. ACN, self-excision cassette including *Cre* driven by the testis-specific promoter. (b) Southern blot analysis to detect double-resistant ES cell colonies using 5' and 3' probes. (c) PCR genotyping of F1 heterozygotes to detect the *Foxc1^{c2}* allele. (d, e) Representative images of lymphatic valves in mesenteric collecting vessels immunostained with

Figure 9 continued

antibodies targeted to FOXC1 and VEGFR-3 from P12 P6 *Foxc1*$^{+/+}$ (d) and *Foxc1*$^{c2/c2}$ (e) mice. Scale bars are 25 µm. (f – h) Representative images of lymphatic valves in mesenteric collecting vessels immunostained with antibodies targeted to FOXC1, FOXC2, and VE-Cadherin from P6 *Foxc1*$^{+/+}$ (f), *Foxc1*$^{c2/+}$ (g), and *Foxc1*$^{c2/c2}$ mice (h). Scale bars are 50 µm. (i – k) Representative images of the mesenteric vasculature immunostained with antibodies targeted to PROX1 and CD31 in P6 *Foxc1*$^{+/+}$ (i), *Foxc1*$^{c2/+}$ (j), and *Foxc1*$^{c2/c2}$ mice (k). Scale bars are 200 µm. (l – n) Representative images of P6 mesenteric vasculature from P6 *Foxc1*$^{+/+}$ (l), *Foxc1*$^{c2/+}$ (m), and *Foxc1*$^{c2/c2}$ mice (n) immunostained with antibodies targeted to FOXC1, FOXC2, and VE-Cadherin show gradual loss of FOXC1 expression in the blood and lymphatic vasculature and smooth muscle cells and conversely the increase of FOXC2 expression in blood vasculature and smooth muscle. Scale bars are 200 µm. (o) Quantification of total lymphatic valve number in lymphatic collecting vessels of P6 *Foxc1*$^{+/+}$, *Foxc1*$^{c2/+}$, and *Foxc1*$^{c2/c2}$ individuals. N = 6 for *Foxc1*$^{+/+}$, N = 8 for *Foxc1*$^{c2/+}$, and N = 8 for *Foxc1*$^{c2/c2}$ individuals. (p) Percentage of mature and immature lymphatic valves normalized to total valves counted in P6 *Foxc1*$^{+/+}$, *Foxc1*$^{c2/+}$, and *Foxc1*$^{c2/c2}$ individuals. Data are presented as mean (± SD) and analyzed using Student's t-test. NS denotes no significance.

morphology (*Figure 9i – k*). To investigate potential differences in valve morphogenesis, we also performed immunostaining of mesentery tissue for PROX1 and CD31 (*Figure 9l – n*). Quantification of PROX1-high valves revealed that total lymphatic valve number was not significantly changed, nor was valve maturation (*Figure 9o and p*). Collectively, these data demonstrate that FOXC2 is able to functionally substitute for FOXC1 transcription activity during lymphatic valve development and maturation.

## Discussion

Mutations associated with the VEGF-C/VEGR3 signaling pathway, including changes in *FOXC2*, as well as key transcription factors involved in LEC specification, such as *Sox-18* and *GATA2*, contribute to the development of primary lymphedema (*Aspelund et al., 2016*; *Jiang et al., 2018*). While recent advances have identified key genes associated with the development of lymphedema, the underlying genetic causes contributing to a majority of cases remains unknown (*Mendola et al., 2013*). In this study, we identify a critical role for FOXC1 during postnatal lymphatic development to regulate cytoskeletal organization in the lymphatic vasculature via regulation of RhoA/ROCK activation and focal adhesion formation. Importantly, we found that high FOXC1 expression is limited to LECs located at the free-edge of the luminal side of valve leaflets, which are exposed to laminar pulsatile shear, as opposed to FOXC2, which is strongly expressed throughout the valve sinuses exposed to reciprocating shear stress (*Sabine et al., 2015*; *Sabine et al., 2016*; *Zawieja, 2009*). Similarly, *in vitro* we demonstrate that LSS induces both FOXC1 and FOXC2 expression, whereas OSS only induces FOXC2. Our findings elucidate a key contribution of FOXC1, complementary to FOXC2, in regulating lymphatic valve maturation and maintenance (*Figure 10*) and provide additional insight into disease processes potentially associated with primary lymphedema.

By utilizing inducible, endothelial-specific loss-of-function genetic mouse models, we show that FOXC2 predominately regulates lymphatic valve maintenance and maturation in comparison to FOXC1. Interestingly, we observed a reduction of FOXC1 expression in collecting vessels of EC-*Foxc2*-KO mice, but no discernable differences in FOXC2 expression were observed in EC-*Foxc1*-KO mice (*Figure 2—figure supplement 1*). This reduction in FOXC1 expression is likely a result of perturbed flow in EC-*Foxc2*-KO mutants due to the degeneration of lymphatic valves, but the maintained expression of FOXC2 in lymphatic valves of EC-*Foxc1*-KO mice may partly explain the differences in severity of the phenotypes observed in *Foxc2* mutants compared to *Foxc1* mutants. However, loss of both *Foxc1* and *Foxc2 in vivo* induces accelerated valve degeneration and vascular remodeling in lymphatic collecting vessels compared to loss of *Foxc2* alone, leading to a rapid onset of chylous effusion from collecting vessels. *In vitro*, both *FOXC1* and *FOXC2* knockdown in LECs resulted in increased actomyosin contractility, which was potentiated by shear stress. However, increased contractility was associated with increased focal adhesion number in *FOXC1*-KD LECs but increased focal adherens junctions in *FOXC2*-KD LECs. RhoA/ROCK signaling has been demonstrated to regulate blood endothelial cell barrier function and permeability by controlling stress fiber formation and focal adhesion dynamics (*Amano et al., 1997*; *Carbajal et al., 2000*; *Cerutti and Ridley, 2017*; *van Nieuw Amerongen et al., 2000*; *Wojciak-Stothard et al., 2001*) and a role for the RhoA/ROCK/myocardin-related transcription factor A (MRTF-A) signaling axis was recently implicated in LEC endothelial-to-mesenchymal transition downstream of TGF-β signaling induction

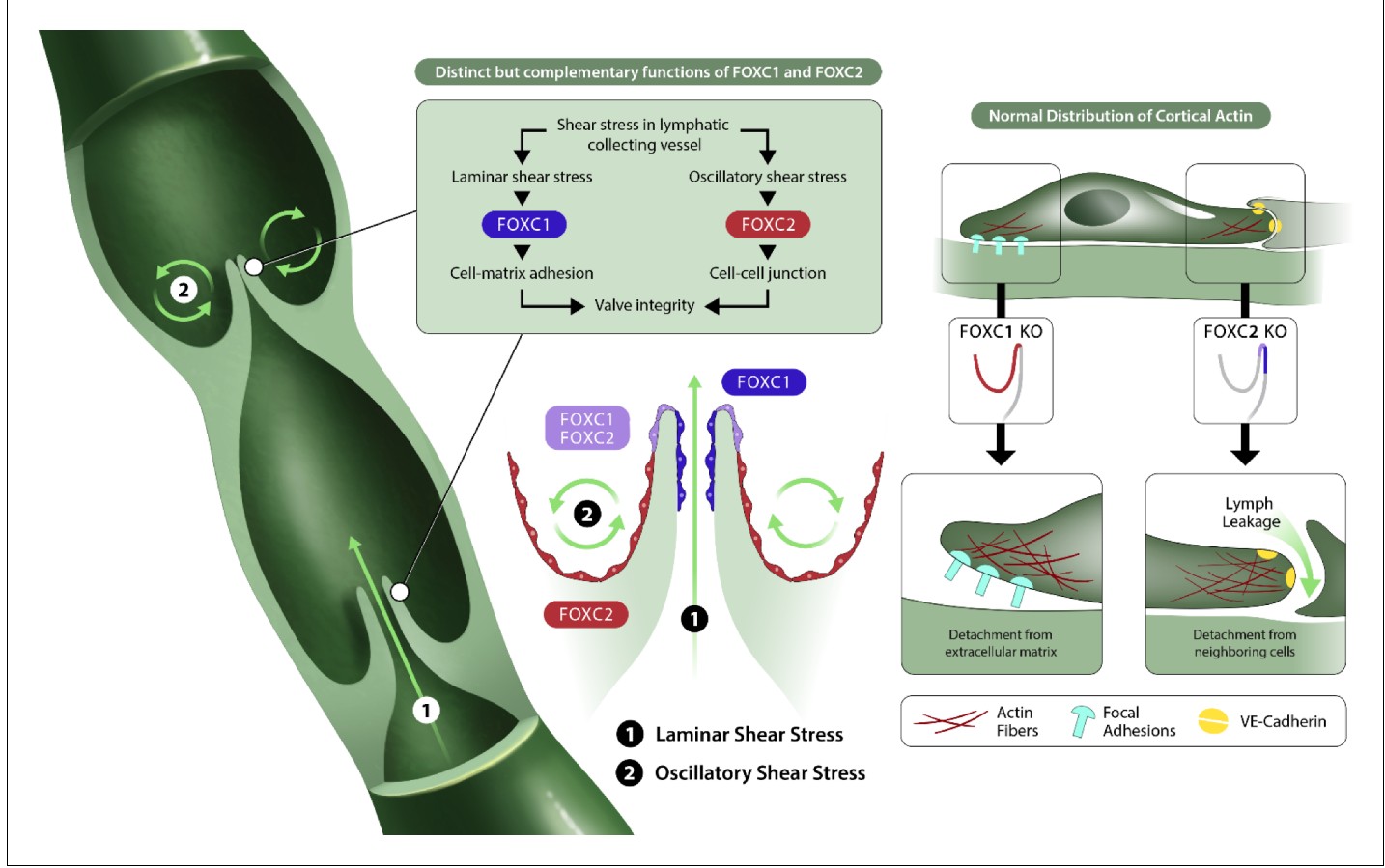

**Figure 10.** FOXC1 and FOXC2 maintain lymphatic valve integrity by regulating cytoskeletal organization in complementary roles. Collecting vessels in the postnatal lymphatic vasculature are characterized by the presence of a high number of intraluminal bi-leaflet valves. These regions are exposed to disturbed flow in the valve sinuses (2) which strongly induces the expression of FOXC2. In contrast, the intraluminal side of valve leaflets is exposed to pulsatile laminar shear (1), which induces FOXC1 in addition to FOXC2. In the absence of FOXC1 and FOXC2, the cytoskeleton undergoes remodeling events in which actomyosin contractility is strongly induced with focal adhesion dynamics perturbed by loss of FOXC1 and intercellular junctions perturbed by loss of FOXC2, ultimately leading to valve degeneration.

The online version of this article includes the following figure supplement(s) for figure 10:

**Figure supplement 1.** Differential expression of genes involved in regulation of focal adhesions in embryonic LECs isolated from the dorsal skin of LEC-specific *Foxc1; Foxc2* mutant mice.

(*Yoshimatsu et al., 2020*). Given increased hypercontractility observed in both *FOXC1*- and *FOXC2*-KD LECs, we hypothesized that RhoA/ROCK signaling was abnormally activated and pharmacological inhibition of ROCK was then able to rescue their perturbed cytoskeletal organization. Recently, ROCK inhibition was shown to reduce chylomicron transport into mesenteric lymphatic vessels by inducing lacteal junction 'zippering' and was proposed as a potential therapeutic for metabolic dysfunction and obesity (*Zhang et al., 2018*). Similarly, we observed an improvement in the reduction of discontinuous cell-cell junctions in mesenteric collecting vessels of both EC-*Foxc2*-KO and EC-*Foxc1; Foxc2*-DKO mice with ROCK inhibition, but degeneration of lymphatic valves was only partially rescued in *Foxc2* mutants. Because lymphatic valve regions were still absent in EC-*Foxc1; Foxc2*-DKO mutants, this suggests that loss of adhesion to the ECM resulting from inactivation of *Foxc1* in addition to impaired cell-cell adherens junctions from inactivation of *Foxc2* strongly reduces LEC valve identity and survivability, which cannot be overcome by inhibition of cytoskeletal reorganization alone. Thus, we propose that FOXC1, in addition to FOXC2, is a key mediator of mechanotransduction where both transcription factors serve distinct, but complementary roles to maintain lymphatic valve integrity by regulating cytoskeletal organization (*Figure 10*).

Our study also identified *PRICKLE1,* a core component of PCP signaling, and the RhoA GAPs *ARHGAP21* and *ARHGAP23* (*Gibbs et al., 2016*; *Zhang et al., 2016*; *Katoh and Katoh, 2003*; *Zallen, 2007*) as targets of FOXC1 and FOXC2 transcription in LECs. Additionally, Prickle1 expression was detected in LECs at the free-edge of valve leaflets (*Figure 5* and *Figure 5—figure supplement 2*), where FOXC1 and FOXC2 are highly expressed. *Prickle1*$^{-/-}$ mutant mouse models were shown to be either embryonic lethal, characterized by loss of apicobasal polarity in epiblast cells (*Tao et al., 2009*), or viable up to P2 with severe skeletal and craniofacial, cardiac outflow tract, atrial septal, neural tube closure, polarity machinery, and actin defects (*Liu et al., 2014*). Intriguingly, *Foxc1* null mutations in mice are also associated with cranial and axial skeletal defects (*Kume, 2009*), and both FOXC1 and FOXC2 are involved in morphogenesis of the cardiac outflow tract (*Seo and Kume, 2006*; *Sanchez et al., 2020*). PCP pathway signaling has previously been implicated in lymphatic intraluminal valve morphogenesis as *Celsr1* and *Vangl2* mutant mice had abnormal mesenteric collecting vessel valves (*Tatin et al., 2013*). Additionally, impaired lymphatic valve formation phenotypes were identified in knockout mice generated from mutations of each member of the ligand-receptor pair *Fat4* and *Dachsous1*, which regulates downstream activation of PCP and Hippo signaling (*Pujol et al., 2017*). Moreover, *FAT4* was recently shown to be a target gene of GATA2 transcription activity and functions in a LEC autonomous matter to regulate cell polarity in response to flow during lymphatic vessel morphogenesis (*Betterman et al., 2020*). Abnormal cytoskeletal organization in *FOXC1*- and *FOXC2*-KD LECs may be explained by reduction of Prickle1 as it was shown to regulate focal adhesion turnover and cytoskeletal organization (*Daulat et al., 2016*; *Lim et al., 2016*; *Zhang et al., 2016*). However, several genes involved in focal adhesion regulation are significantly downregulated in LEC-*Foxc1; Foxc2*-DKO mice including *Actn1, Itgb3, Rras, Src, Tln1, Tln2, Tns1, Tns3,* and *Vcl* that may be related to the phenotype observed in LECs with inactivated *FOXC1* and *FOXC2* (*Figure 10—figure supplement 1*). Whether Prickle1 is directly involved in regulating lymphatic valve morphogenesis and other aspects of lymphatic vascular development is unknown and future studies investigating its role and other regulators of cytoskeletal activity in the lymphatic vasculature should be of focus.

By utilizing a transgenic mouse model in which the *Foxc2* coding region was recombined into the *Foxc1* locus, we also demonstrate that FOXC2 is able to functionally substitute for FOXC1 during lymphatic vascular development. Both FOXC1 and FOXC2 share nearly identical DNA binding domains (97% identity; 99% similarity) (*Kume, 2009*) and *FoxC1* and *FoxC2* genes were likely to have been generated from the duplication of the ancestral *FoxC* gene in deuterostomes (*Carlsson and Mahlapuu, 2002*). Although both FOXC1 and FOXC2 share cooperative roles in regulating development of several organ systems, our evidence demonstrates that response to differing mechanical stimuli is key for the induction of FOXC1 or FOXC2 expression in LECs to control their separate functions. The response of LECs within collecting vessels to differing shear stress forces may explain the differences we observe between our *in vitro* results where knockdown of *FOXC1* strongly induces actomyosin hypercontractility and increased focal adhesion number, yet inactivation of *Foxc1 in vivo* only results in a modest phenotype and we observe that FOXC2 can functionally substitute for FOXC1. Our group previously reported that loss of *FOXC2 in vitro* leads to distinct phenotypes between LECs under OSS and LSS. *FOXC2* inactivation and OSS induced the formation of circumferential thick actin fibers and strong disorganization of adherens junctions whereas LSS enhanced cell elongation compared to cells under static and OSS culture conditions (*Sabine et al., 2012*). Therefore, although FOXC1 and FOXC2 have the same DNA-binding capacity, their transcriptional targets are likely to be different in cells under OSS or LSS.

How LECs are able to sense differences in flow forces remains poorly understood. Most work to date has focused on how blood endothelial cells sense and respond to flow forces and temporal dynamics, identifying proteins expressed at the cell-surface such as VE-Cadherin, CD31/PECAM-1, VEGFR-2, and VEGFR-3, G-protein coupled receptors, and sphingosine 1-phosphate receptor one as components of a signaling pathway that senses and converts the mechanical stimulus provided by shear stress into downstream activation of pathways such as VEGFR and PI3K signaling (*DePaola et al., 1992*; *Tzima et al., 2005*; *Chachisvilis et al., 2006*; *Jung et al., 2012*; *Conway et al., 2013*; *Baeyens et al., 2015*; *Coon et al., 2015*; *Sun et al., 2016*). In regard to the lymphatic vasculature, loss of *Pecam1* or *Sdc4* and LEC-specific deletion of *Cdh5 in vivo* resulted in impaired valve morphogenesis phenotypes and poor orientation and elongation of LECs with the direction of flow in mesentery collecting vessels (*Wang et al., 2016*; *Yang et al., 2019*). Recently, an

*in vitro* study utilizing microfluidic chambers implicated the role of E-selectin, an adhesion molecule highly expressed in ECs, in LEC response to shear stress gradients and their ability to reorganize perpendicularly to the direction of flow and increase expression and nuclear localization of FOXC2 (*Michalaki et al., 2020*). However, E-selectin knockout mice were not reported to have lymphatic dysfunction *in vivo* (*Frenette and Wagner, 1997*), suggesting a possible functional redundancy with other cell-surface proteins during lymphatic development. Whether these cell-surface proteins are responsible for the differences in FOXC1 and FOXC2 induction in response to LSS and OSS is not known and further investigation into how shear stress regulates differential expression of downstream FOXC1 and FOXC2 targets will be of critical importance for the focus of future studies.

In conclusion, our findings identify a key role for FOXC1, in addition to FOXC2, in controlling postnatal lymphatic valve maturation and maintenance processes as key mediators of mechanotransduction to control cytoskeletal organization and RhoA/ROCK signaling activity in valve LECs. Our results not only advance the current understanding of molecular mechanisms contributing to valve morphogenesis in the lymphatic vasculature, but also identifies RhoA/ROCK as a potential therapeutic target for treatment of hereditary forms of lymphedema.

## Materials and methods

### Animal models

*Foxc1^{fl/fl}*, *Foxc2^{fl/fl}*, *Foxc1^{fl/fl}; Foxc2^{fl/fl}*, *Cdh5-Cre^{ERT2}*, and *Prox1-Cre^{ERT2}* mice were described previously (*Sasman et al., 2012*; *Sörensen et al., 2009*; *Srinivasan et al., 2007*). Tamoxifen dissolved in corn oil was orally administered (75 or 100 µg per neonate) from P1 to P5 using techniques as previously described (*Butchbach et al., 2007*) with slight modifications. In some experiments, mice were injected subcutaneously with DPBS (10 µL/g) or Y-27632 (Fisher, #12-541-0, 15 µg/g) 1 to 2 hr after administration of Tamoxifen from P2 to P5.

Generation of *Foxc1^{c2/c2}* mice is summarized in *Figure 9*. Using overlapping *Foxc1* genomic clones isolated from a 129/SvJ mouse genomic library (*Kume et al., 1998*), the targeting vector consists of a 4.5 kb 5′ homology region and 4.1 kb 3′ homology region. The *Foxc1* coding region was replaced with the cDNA coding (from the start codon to the stop codon) for *Foxc2* by using restriction enzyme digestion and PCR cloning. To facilitate the elimination of the selectable marker gene *Neo^r* in mice, we introduced the ACN cassette flanked by two flox sites (*Bunting et al., 1999*) in the targeting vector, which was inserted in the 3′ untranslated region of *Foxc1*. The ACN cassette codes for *Cre* driven by the testis-specific promoter of the *angiotensin-converting enzyme* gene and for *Neo^r* driven by RNA polymerase II; thus the entire cassette is deleted as it passes through the male germ line of mice (*Bunting et al., 1999*). For negative selection, a PGK-TK cassette was placed at the end of the 3′ homology region. The resultant targeting vector was confirmed by sequencing.

Sall-linearized targeting vector (100 µg) was then electroporated into TL1 ES cells (129S6) as described (*Kume et al., 1998*). The cells were selected with G418 and gancyclovir. Double-resistant ES cell colonies were screened by Southern blot using 5′ and 3′ probes. One targeted clone (7F11) was injected into host (C57BL/6) blastocysts and produced germline chimeras. Targeted ES cells were screened by Southern blot using 5′ and 3′ probes following BamHI digestion (*Figure 9b*). Chimeras were mated with Black Swiss (Taconic) females and maintained by interbreeding on a mixed (129 x BlackSwiss) genetic background. After germline transmission, F1 heterozygous knock-in mutant (*Foxc1^{c2/+}*) mice (*Figure 9c*) were then intercrossed to generate *Foxc1^{c2/c2}* mice. Offspring were genotyped either by Southern blot using the 3′ probe or PCR using two specific primers: 5′-CAGCGTTTTCTGCAAAACATA-3′ and 5′-GGACACATAGGCTGATCTCCA-3′. Weaned mice for continued animal colony maintenance from several breeding pairs were generated at near Mendelian ratios for *Foxc1^{+/+}* (77/284, 27.113%), *Foxc1^{c2/+}* (125/284, 44.014%), and *Foxc1^{c2/c2}* (82/284, 28.873%) individuals.

Genotyping of mice for use in analysis was performed by Transnetyx Inc (Cordova, TN) using real-time PCR.

### Mouse tissue collection, staining procedures, and image acquisition

Dissection, immunostaining, and imaging of the lymphatic vasculature in mesentery tissue from 4 week old C57Bl6 mice for whole-mounts was performed as previously described (*Sabine et al.,*

*2018*). Briefly, dissected mesentery tissue was fixed with 4% PFA, washed with PBS, followed by incubation with 10–20% sucrose solutions, then washed again in PBS before incubation with blocking buffer containing 0.5% BSA, 5% donkey serum, and 0.5% Triton X-100 in PBS, immunostained with primary antibodies at room temperature for 24 hr, washed with 0.5% Triton X-100 for 24 hr, followed with incubation with secondary antibodies for another 24 hr, washed again, rinsed with PBS and cleared for 2 days in 88% Histodenz (Sigma, D2158) in RIMS buffer, and eventually mounted in fresh Histodenz solution.

The lymphatic collecting vessel vasculature of neonates was analyzed by whole mount immunostaining of mesentery tissue harvested from pups at the indicated time. Briefly, mesentery tissue was dissected from the intestinal tract, laid out in plastic dish and left until it was firmly attached. Following fixation with 2% PFA in PBS, tissues were washed with PBS, then permeabilized and blocked in PBS solution containing 0.5% BSA, 5% serum, 0.3% Triton X-100, and 0.1% Sodium Azide. Tissues permeabilized/incubated with blocking buffer were then incubated with primary antibodies listed in *Table 1* overnight followed by washes prior to incubation with Alexa Fluor conjugated secondary antibodies and dyes listed in *Table 1* overnight. After subsequent washes, samples were post-fixed in 4% PFA, washed in PBS, and flat mounted on slides in mounting medium. Whole-mount staining images are shown with the same orientation, i.e., with flow in the direction of bottom-to-top.

**Table 1.** Antibodies and Dyes.

| Antigen | Reactivity | Host Species | Origin |
|---|---|---|---|
| **Primary antibodies** | | | |
| Active caspase 3 | Human/Mouse | Rabbit | R and D AF835 |
| **α**9-integrin | Mouse | Goat | R and D AF3827 |
| CD31 | Mouse | Rat | BD Pharmingen 553370 |
| Foxc1 | Human/Mouse/Rat | Rabbit | Cell Signaling 8758S |
| Foxc2 | Mouse | Sheep | R and D AF6989 |
| Foxc2 | Human/mouse | Rat | Kind gift from Dr. N Miura (*Miura et al., 1997*, *Genomics*) |
| Paxillin | Human/mouse | Mouse | BD Transduction Clone 349–610051 |
| Phospho-MLC2 (Thr18/Ser19) | Human | Rabbit | Cell Signaling #36745 |
| Prox1 | Human | Goat | R and D AF2727 |
| Prickle1 | Human | Rabbit | Thermo Fisher PA5-51570 |
| VE-Cadherin | Mouse | Rat | BD Pharmingen 555289 |
| VE-Cadherin | Human/Mouse | Goat | R and D AF1002 |
| Vegfr3 | Mouse | Goat | R and D AF743 |
| Vinculin | Human | Mouse | Sigma – Clone hVIN-1 V9131 |
| **Secondary Antibodies** | | | |
| Alexa 405-conjugated | Rat | Donkey | Abcam ab175670 |
| Alexa 488-conjugated | Rabbit/Rat/Sheep Goat/Mouse | Donkey | Thermo Fisher |
| Alexa 555-conjugated | Goat/Mouse/Rabbit Rat | Donkey | Thermo Fisher |
| Alexa 568-conjugated | Goat/Rat | Goat/Donkey | Thermo Fisher |
| Alexa 647-conjugated | Goat/Mouse/Rabbit Rat | Donkey | Thermo Fisher |
| **Dyes** | | | |
| Hoechst 33342 | - | - | Thermo Fisher |
| Alexa 488-conjugated phalloidin | - | - | Thermo Fisher |
| DAPI-containing Prolong Gold antifade reagent | - | - | Thermo Fisher |

**Table 2.** Primers used for qPCR analysis.

| Gene | Forward | Reverse |
|------|---------|---------|
| Foxc1 | TTCTTGCGTTCAGAGACTCG | TCTTACAGGTGAGAGGCAAGG |
| Foxc2 | AAAGCGCCCCTCTCTCAG | TCAAACTGAGCTGCGGATAA |
| Ppia | CAAATGCTGGACCAAACACA | TGCCATCCAGCCATTCAGTC |
| 18S | GAAACTGCGAATGGCTCATTAAA | CCACAGTTATCCAAGTAGGAGAGGA |

## Imaging

Imaging was performed using a Zeiss AxioVision fluorescence microscope and Zeiss Axiovision software, Zeiss LSM-510 Meta, LSM-800 and LSM 880 confocal microscopes and Zeiss Zen Blue acquisition software, or using a Nikon A1 Confocal Laser Microscope System NIS-Elements software. Images were processed with Imaris and Adobe Photoshop software. Imaris colocalization function was used to produce pictures showing p-MLC2 (*Figure 4b*), vinculin (*Figure 4c*) or paxillin (*Figure 4—figure supplement 1b*) staining that is associated with F-actin staining.

## RNA isolation and qPCR analysis

Hearts from neonatal mice were digested in collagenase Type I solution (2 mg/mL) for 40 min at 37°C with gentle agitation. Cells were then filtered through a 70 μm cell strainer and the pellet was resuspended in Buffer 1 (PBS, 0.1% BSA, 2 mM EDTA, pH 7.4). The cell suspension was then incubated with magnetic Dynabeads (Invitrogen) pre-coated with CD31 antibody to isolate the endothelial cell population. After several washes with Buffer 1, RNA was extracted from endothelial cells using RNA STAT solution (Tel-Test) followed by phenol-chloroform treatment. Extracted RNA was subjected to DNAse I treatment and concentration was determined using a NanoDrop machine (Thermo Scientific). cDNA was synthesized using an iScript reverse transcriptase kit (Bio-Rad) according to the manufacturer's instructions. Triplicates of cDNA samples for qPCR analysis were performed using a Fast qPCR machine (Applied Biosystems), Fast SYBR reaction mix (Applied Biosystems), and gene specific primer sets. Peptidylprolyl isomerase A (*Ppia*) or 18S was used as an internal standard for mRNA expression. Primer sequences are provided in *Table 2*.

## Cell transfection and immunostaining

Culture of human intestinal LECs was performed as described previously (*Sabine et al., 2015*). LECs were fixed using 4% PFA in PBS, then permeabilized with 0.1% Triton X-100 followed by blocking with 5% donkey serum and incubation with primary and secondary antibodies listed in *Table 1*. Knockdown experiments were performed by transfecting cells with 40 nM siRNA using Lipofectamine RNAiMAX (Invitrogen). siRNAs are listed in *Table 3*. Knockdown efficiency was confirmed by immunostaining (*Figure 3—figure supplement 1d,e*). LECs were analyzed after 48 hr, or used 24 hr post-transfection for flow experiments. For rescue experiments, transfected LECs were treated with vehicle control or 10 μM Y-27632 (StemCell, 72304) diluted in PBS every 12 hr with analysis at 48 hr. No cell lines from commonly misidentified cell lines were used. Primary lymphatic endothelial cells were isolated and cultured as described in *Norrmén et al., 2009*. Cell identity was confirmed by staining for lymphatic endothelial cell marker PROX1. Cells are isolated and cultured in the presence

**Table 3.** List of siRNAs.

| Gene | Species | Company | Reference (Sequence) |
|------|---------|---------|----------------------|
| Control | Human | QIAGEN | AllStars Neg. Control siRNA-1027281 |
| FOXC1 | Human | Origene | SR320173<br>#1 -rGrArUrArArArArCrArCrUrArGrArArGrUrUrArCrCrUrATT<br>#2 - rCrUrArGrUrCrCrArUrGrUrCrArArArUrUrUrUrArCrUrAAA |
| FOXC2 | Human | Thermo Scientific (Dharmacon) | FISSH-000119 (AGGUGGUGAUCAAGAGCGAUU)<br>FISSH-000321 (CAACGUGCGGGAGAUGUUCUU) |

**Table 4.** Primers used for ChIP analysis.

| Evolutionary Conserved Region (ECR) | Forward | Reverse |
| --- | --- | --- |
| *PRICKLE1* ECR-1 | ACACAAGGCGGTGCTCTAAT | CTTGTTTCAAATGGGTGCT |
| *PRICKLE1* ECR-2 | GCAAATGGCACATTTAAGCA | TGGCTCCTTTTCTTTGCTGT |
| *PRICKLE1* ECR-3 | AGGCAGACCCTTTTTGGAAT | GGAAGCTTGCAACTGTCTCC |
| *PRICKLE1* ECR-4 | GCAAGTGTGCAAACCCTTAAC | CAGCTGGAGCCTGAAGAAAG |
| *PRICKLE1* ECR-5 | CCACCAGACAGCAAGATGAA | TTGACCGTCCCCAACATTAT |
| *PRICKLE1* ECR-6 | TGCCTTGTTCATGGTCTCAG | AAGAAAAACAAACGGCATCG |
| *ARHGAP21* ECR-1 | GCTTGCTAGCCAAGGACAAG | CCTACCTGCAACCTGGTGAT |
| *ARHGAP21* ECR-2 | ATCACCAGGTTGCAGGTAGG | GGCAGAACTGTAGGTTTACATTTAG |
| *ARHGAP21* ECR-3 | TGTGGAAGGCCATTCTATGA | GTTTTGCAAAGGCTTCAACC |
| *ARHGAP23* ECR-1 | CCTCCCTGCTCCTAAGTTGA | CCAAGTCTTTCAGCCCTGTC |

of 30 μg/ml Gentamicin and 15 ng/ml Amphotericin B, cultured for maximally 10–12 passages and tested negative for mycoplasma contamination by DNA staining.

### *In vitro* flow experiments

Flow experiments with cultured LECs were performed as described previously (*Sabine et al., 2015*). Briefly, LECs were seeded at confluence on slides (μ-Slide I$^{0.8}$ Luer; Ibidi) coated with 40 μg/ml human fibronectin, cultured for 24 hr and then subjected to LSS (4 dyn/cm$^2$), OSS (4 dyn/cm$^2$ and flow direction change every 4 s) using Ibidi Pump system, or kept in static conditions for an additional 24 hr prior to fixation, immunostaining, and mounting using Ibidi Mounting Medium (ibidi GmbH).

### Forkhead box C transcription factor binding prediction analysis

Putative FOX-binding sites were determined first by using the Hypergeometric Optimization of Motif EnRichment (HOMER) (*Heinz et al., 2010*) suite of tools to scan the entire Genome Reference Consortium Human Build 37 (GRCh37 or hg19) genome corresponding to the conserved RYMAAYA FOX transcription factor binding motif. The output file was then uploaded to the UCSC genome browser (*Kent et al., 2002*) to identify putative binding sites corresponding to transcriptionally active areas as indicated by histone modification, DNAse sensitivity, and additional transcription factor chromatin immunoprecipitation data as per work reported and summarized on the Encyclopedia of DNA Elements (ENCODE; https://genome.ucsc.edu/ENCODE/)(2012). Putative sites in the human genome were then searched against the mm10 mouse genome using the ECR Browser (https://ecr-browser.dcode.org) (*Ovcharenko et al., 2004*) and rVista 2.0 tools to identify conserved and aligned putative binding sites between mouse and human sequences.

### ChIP assay

Human dermal lymphatic endothelial Cells (HDLECs) from juvenile foreskin (Promo Cell, #C12216) (https://www.promocell.com/product/human-dermal-lymphatic-endothelial-cells-hdlec/) were cultured and used according to the manufacturer's protocol. The cells were cross-linked with 1% formaldehyde, followed by sonication. The sheared chromatin was immunoprecipitated with dynabeads (Invitrogen, #10004D) conjugated with anti-FOXC2 antibody (Abcam, ab5060), anti-FOXC1 antibodies (Origene, TA302875 and Abcam, ab5079), or control IgG (Thermo Fisher Scientific, # 02–6202). DNA extraction and PCR were performed as previously described (*Fatima et al., 2016*) with primers listed in *Table 4* targeting identified evolutionary conserved regions (ECRs) containing putative binding site sequences shown underlined listed below. Images were acquired with a ChemiDoc Touch Imaging System (Bio-Rad) and band intensities were analyzed with Image Lab software.

*PRICKLE1 ECR-1.*>hg19 chr12:42876176–42876523
ACCACACAAGGCGGTGCTCTAATGAGCCCATTATTTTCCATAATGGGGGATGCAGAT
ATTTTCTCAAAATCGTGTTCTCCTCAGTCTTCTATTGATTTTTTGGATTTCTATTTTCAA
CAGTGGCCCGAGGAAACGGCAGCCAGACTTGACTCCAATGTACACACAGACTCAG

GTTTCGCCCCGTCACC**TGTTGGCT**CCTGAACAACACTCCTCTTTGTGAAACTTACA
GCACCCATTTGAAACAAGTTTCCAGAGAAAACACTTCAAGAAAGTGTTTGAGAAGT
CACAAGACTCGAGTTGTAAAAACAAATTCCACACATAGCCTGGCTTATAAGGACACA
GACTAAC
*PRICKLE1 ECR-2.*>hg19 chr12:42878066–42878454
GAGCACCTACCGCCGCCCGCCCGCTCCATTCTCCCGAGCCCAGTGAGTGAAGCC
GCTAAGATGCAAATACCCTAGGACGCTTATGTAAACTTCCCCCTCCCGCAGGTGCA
CGCGCGGGCCACGAAACGCTGGGAGAATATGAAAGGCCACCTCTTAAAGAAATCAT
CTCCACTCTGCCCATAACAATGATGTCAGCAAATGGCACATTTAAGCAAGTTCTCAC
TTAGAAGGGCTCATTAGCATATGAATTCTCTTAGGACTTTCCCTGCATTTCGGAGTGA
TTCCTACTGCTTAGCGCAGGAGATTTATTTTTATCAGTAAATAACAGCAAAGAAAGG
AGCCAGGTCACGCGATGTACTAACTCAAGTACACTACTGAGAGTTTTTAC
*PRICKLE1 ECR-3.*>hg19 chr12:42879624–42879943
AGTTACTTGTAAATACGTTTTGTTATATTTTCAACAGGTACTGTCATGGTTTATTACCC
ATTGTAAGCGTTTTTAGTGATGAAGCAGTGTGGAGACAGTTGCAAGCTTCCTACAAA
GGTTCAATCTCAATGAAACAGACTTTAGTCTGGTCTGAAAGGGGTTGTTATTGTCAA
GGGTGAACTTATGCAGAATGGAGAGCAAGGCCCCCAACCAGCATCCTTTTGTTTCA
GCCAGGTGGAATATTCATGCTTGCAGATCACACTTTAGGGGCCAGTTGAGAAAGGA
AGCCCAAAAATTCACAGGGCTTGGTTCCCTCGCTC
*PRICKLE1 ECR-4 and ECR-5.*>hg19 chr12:42981201–42982292
TACATTAGCCATGACTTATTAAACTTGGAGATTTTCAAGTTCATCAGCAAGTGTGCAA
ACCCTTAACTGTGGGTAACATCCATTTATTTGTAGCACCTTTCGGTTTTAATATGTAGA
GCACATGCATTGTTAACCTCTAAATCCTTTGT(ATAAACA)TTTCTGGAAGAGCTGGT
AAAATATCTCCTTCTGTGTTTCCTGACTCGCCAGTTGATGGCATTTAGAAACCCTCT
GGTACCAGCAGGTGCTGTATTTTGCTTTCTTCAGGCTCCAGCTGGGCTACAATGAC
AGATTCCTGTCCCAGGCCAAGCCTAGCCACCAAGGCTAGGACCACATTGGAGGCA
AACTGAACCAGGCTCCACCAGACAGCAAGATGAATGGCTGCTGTTTAAGTTTAAAAT
CCCCTGGTGGGA(GTAAATATTGTTCCAGAGAAAAGCCTTGACAAATA)CTGCGTCA
TCCTTACAGAACTGTCTTGATTAAAGCAGAATCTTTGGATTAAGTTGATGCTCAATTC
AAAATGTATCTATCTTGCTGTCATGGGATTTTTTTTTGTTTTTTTTTTCCTTTCTAGAGT
CTGAAAACAGACATAATGTTGGGGACGGTCAAACAAGGCTGCCGGCTCCCAAGGG
GCTAGAGTCCACTCCTGATAATAGAAGGCGGCTGAACACTGACACTTCACTGAGGA
TAATGGAGACAGCAAAGGCTTAGTGGGAAAGGGCCAGTTGTCACCTAAGTGACAG
GCAACAGCTGAGCTCACACATCTGGAGCCGGACTACGGCAAACATTAGCAACCCTC
ACCAGTCTACACCTTGGGCCTGTCTGAAAAGACAGATGGAAGTTCCCTCTACTCCT
AAAGTACATTAAAAAATGTCTGATGGTGAACCACATCAATTATATAACATCAACTGCAG
GCACAGCCTTCCAAAGTACTGATTAAGACGAGGCAGTAGACAACACTGTATGCATG
AACAGATACAAGATACCATTTCAGTGATTTGTCATTCATAAAACTTATCCTAAAAGACA
CATATACATGCATCCATTTGATAGCACAAATGCATGTTAACTCTGCAGGAGAGGCAGA
TTTTTACATGT
*PRICKLE1 ECR-6.*>hg19 chr12:42982328–42982743
CCTTTAGGCTGGTTCCCGTGGTGTGTTTGCCTTGTTCATGGTCTCAGTTCTGCCGC
TGATACCCTTTTAAAAATCAGCAACCAAACGCGTTCGGCTTGTGATCCTGAACCCCC
TTAGGCAAGCTGGAACTAAGCGTGATGCAGCCGTCCTCCCTCTCTCCCAACCCCCA
ACCTCGTTCTTCAGCCTCCTGAAGACAATCTGTGAACAATTTTCCCAAAGTCCCAAG
AATAACACAGCACTGCCAATAGTCACTGGCGATGCCGTTTGTTTTTCTTAGAGGGTA
ATGAAAATTTAACAGCTTTCTGCTGCATCCTGAGTCCCGCTCCTAATAACTATTAACA
TGCCTAGTTTCTTCAACTTTTTCTACCTCAAGAGAGGAAGACGCTCCCATTTTTTCC
CTATATCTGTGCTACAT
*ARHGAP21 ECR-1 and ECR-2.*>hg19 chr10:25009076–25009514
CTGTCATTGTTAATAAAAGCCAAGTGTGCAACAAACTGGAAATACTGCTTGCTAGCA
AGGACAAGATGTGTCTAAATTCTTGGTTTCAGGACATCTCTTAAATGACCAAAAAAAA
AAAAAAAAATCAAATAACATTAGTTCTT(GCAAACAGAATGCAAATACAATGCTAATTA
AAGTATTCACAAGACAATGACAAATA)AGGCTTCAGGACCACAATACATATTATTATGT
AACTGCAATACACATTAAGCAATCACCAGGTTGCAGGTAGGCCTTCCAAAAGGAGTT
ATTATGGTTTACCGTGATCAGAGGATTGTGGTGTTCCACTTAATCATGCTTTTGCCTG
CAAGCAGGTGTTTACAGATGTCAAAAGTAAAACACTGATTCTAAATGTAAACCTACA
GTTCTGCCTAATAAATTGTACAGTAATAGCACAC
*ARHGAP21 ECR-3.*>hg19 chr10:25017295–25017667
CTATATTAATAAAAACTACAAGAAAGCTTTATACACTAAATCTAGGCAAGACATTTATGA
AGATGAGAACTGTATCCTTAAAAGGTAAGTGTTGGCTTTGCTAATGACATAATATTGT

TTTGGTGAACCAATATCAAGGGAAAAAATGTCAAAGCCAAAAATAGAGGCAAAGTAT
CCCAGCCCCTGGTGTGGAAGGCCATTCTATGATAATCTATGAATGATTTCTACTCTGA
ATATGTTAACAGAAGCTGGCACATCTGAGAAGCACAAGTGTTTGCTAGTGAATCCAC
AAATGAAATTTGCAATTTGGGTTGAAGCCTTTGCAAAACTACGTTAAGACCAATAGC
CCTCAGAAGAGTAAGGGGTTTTGTTT
*ARHGAP23 ECR-1.*>hg19 chr17:36585363–36585767
TTCTAGACACAGGCCCAGGACCCCGGGCTCTGCCGGCGAGGCTGCCCTCCCCTC
TGCCCTCTCCGACCGGCTGTGGGTGGGTCAGAGCGCGGGGTGCCAGGGGCATTA
CTCAGCGCTGGGCTGCTCTGCCTGGGTTCTTTCATCTGCCAGCTGCTGAGGCTGG
GGAGGGGCCAGCAGGGGCCTCCCAGCCCCATCCCCCCATCAGGGCCATTCCCTTA
CCTCTGAGCCTGGCTGCCCGCCCTGCAGGAGCCCCCCAGCAGGCCTCCCTGCTC
CTAAGTTGAAGGGTTGAACACTGTCAGGCCAACAGTTTCCCTGAGCTCGGAAAAG
AAATTCCCCGGGGTCCAGGTTGAGGTCAAGGCCAGGGCTGAGGCCTGTTCCTCTT
TAGACAGGGCTGAAAGACTTGGG

## Quantification

To identify lymphatic valves, mesentery tissue was stained with PROX1 antibody and areas of high expression were quantified as mature (visible leaflets) or immature (no visible leaflets) on four lymphatic collecting vessels per individual. Total number of valves were determined and percent of mature valves was normalized to the total counted per individual. For assessment of apoptosis in lymphatic collecting vessels, mesentery tissue was immunostained with PROX1 and active caspase-3 antibody and the percentage of PROX1/caspase 3-positive LECs was quantified from 20X high-power fields generated from confocal z-stacks using Fiji software to determine the total number of LECs per field using thresholded PROX1 immunostaining. Quantification was completed from three biological replicates.

For *in vitro* quantifications, relative nuclear intensity levels were measured using Fiji software. Nuclei were considered as regions of interest from thresholded Hoechst staining pictures. FOXC1 or FOXC2 intensity was then measured in each nuclei using RawIntDen function. For quantification of F-actin and p-MLC2 area per cell, LECs were fist manually segmented and defined as individual regions of interest, then F-actin (or p-MLC2) staining was thresholded using similar parameters for all pictures and F-actin$^+$ (or p-MLC2$^+$) area was measured per each region of interest. Quantification was completed from three independent experiments.

## Statistics

Statistical analysis for *in vivo* experiments was performed using GraphPad Prism v5 or v8. *P* values were obtained by performing a 2-tailed Student's *t* test or one-way ANOVA and Tukey's test. Data are presented as mean ± standard deviation of representative experiments from at least three biological replicates. For quantification of apoptosis in LECs of compound *Foxc1; Foxc2* mutant mice, the ROUT method with Q set to 1% was used to identify outliers in the data set that were then excluded from statistical analysis using student's t-test. *P* values less than 0.05 were considered statistically significant. Statistical analysis for in vitro experiments was performed using GraphPad Prism v8 to perform mixed-effects analysis and calculation of correlation coefficients. *P* values less than 0.05 were considered statistically significant for mixed-effects analysis.

## Study approval

All procedures and animal studies were approved by Northwestern University's IACUC or by the Animal Ethics Committee of Vaud, Switzerland.

## Acknowledgements

We thank Justine Epiney for technical assistance and Lauren Kalinoski for providing illustrative and graphic design work for the development of *Figure 10*. Imaging work for neonatal mouse tissues was performed at the Northwestern University Center for Advanced Microscopy generously supported by NCI CCSG P30 CA060553 awarded to the Robert H Lurie Comprehensive Cancer Center. This work was supported by the NIH (R01HL126920 and R01HL144129 to T Kume, 5T32HL094293 to PRN).

## Additional information

### Funding

| Funder | Grant reference number | Author |
|---|---|---|
| National Institutes of Health | R01HL126920 | Tsutomu Kume |
| National Institutes of Health | R01HL144129 | Tsutomu Kume |
| National Institutes of Health | 5T32HL094293 | Pieter R Norden |

The funders had no role in study design, data collection and interpretation, or the decision to submit the work for publication.

### Author contributions

Pieter R Norden, Amélie Sabine, Conceptualization, Data curation, Formal analysis, Investigation, Methodology, Writing - original draft, Writing - review and editing; Ying Wang, Data curation, Formal analysis, Investigation, Methodology, Writing - review and editing; Cansaran Saygili Demir, Ting Liu, Formal analysis, Investigation, Methodology; Tatiana V Petrova, Conceptualization, Supervision, Writing - review and editing; Tsutomu Kume, Conceptualization, Supervision, Writing - original draft, Project administration, Writing - review and editing

### Author ORCIDs

Pieter R Norden (iD) https://orcid.org/0000-0002-5960-5245
Amélie Sabine (iD) http://orcid.org/0000-0002-7512-6703
Ying Wang (iD) https://orcid.org/0000-0002-7852-386X
Tsutomu Kume (iD) https://orcid.org/0000-0002-6005-5316

### Ethics

Animal experimentation: All procedures and animal studies were approved by Northwestern University's institutional animal care and use committee (IACUC) (Protocol: IS00008794) or by the Animal Ethics Committee of Vaud, Switzerland.

### Decision letter and Author response

Decision letter https://doi.org/10.7554/eLife.53814.sa1
Author response https://doi.org/10.7554/eLife.53814.sa2

## Additional files

### Supplementary files

• Transparent reporting form

### Data availability

All data generated or analysed during this study are included in the manuscript and supporting files.

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
