## [Decision Letter]

**Acceptance summary:**

This study provides new insight into the mechanically regulated activation of, and distinct roles for, FOXC1 and FOXC2 in lymphatic vessel valve development. The identification of unique roles for these closely related transcription factors is intriguing and will pave the way to a deeper understanding of the transcriptional mechanisms regulating their distinct activities. The study also implicates FOXC1 as a gene that may potentially underlie primary lymphoedema.

**Decision letter after peer review:**

Thank you for submitting your article "Shear stimulation of FOXC1 and FOXC2 differentially regulates cytoskeletal activity during lymphatic valve maturation" for consideration by *eLife*. Your article has been reviewed by three peer reviewers, including Natasha L Harvey as the Reviewing Editor and Reviewer #2, and the evaluation has been overseen by Didier Stainier as the Senior Editor. The following individual involved in review of your submission has agreed to reveal their identity: Mark Kahn (Reviewer #3).

The reviewers have discussed the reviews with one another and the Reviewing Editor has drafted this decision to help you prepare a revised submission.

Summary:

Here, Norden, Sabine and colleagues dissect the roles of transcription factors FOXC1 and FOXC2 in lymphatic vessel valve development, revealing that FOXC1 primarily controls focal adhesions in lymphatic endothelial cells, while FOXC2 primarily controls adherens junction integrity. The work builds on previous studies by the authors that have demonstrated (1) the importance of FOXC2 and shear stress for lymphatic vessel valve development and (Sabine et al., 2012, Sabine et al., 2015), (2) important roles for both FOXC1 and FOXC2 in embryonic lymphangiogenesis via regulation of RAS/ERK signalling (Fatima et al., 2016). Aspects of this current study overlap with some of the authors' prior work and this should be clearly articulated in the manuscript where relevant. Distinctions and extensions of the work should also be clearly articulated. There are key novel and interesting findings described in this manuscript that address important questions regarding the distinct mechanisms by which FOXC1 and FOXC2 activity are regulated, together with the mechanisms via which these transcription factors regulate cytoskeletal architecture.

Essential revisions:

1) The phenotype caused by early postnatal deletion of *Foxc1* seems modest, and is characterized by a larger proportion of immature valves in mesenteric lymphatic vessels. It thus becomes important to know how the authors define immature valves. It is not obvious why those shown in Figure 2E, F are defined as immature, and not representing valves that are imaged from a different (90°) angle compared to valves that show the typical v-shape. Staining of the valve leaflets would demonstrate this more clearly. The inclusion of staining for markers of valve maturation such as α5-laminin and/or α9-integrin would help to demonstrate this point. To more convincingly demonstrate that *Foxc1* modulates the *Foxc2* KO phenotype, side-by-side comparison of *Foxc2* single and *Foxc1*;*Foxc2* double mutants should be shown. Of note, deletion of *Foxc2* seems to be incomplete (Figure 1—figure supplement 1). Considering that the authors show that *Foxc2* can compensate for *Foxc1* function, *Foxc1* deletion in combination with the incomplete *Foxc2* deletion may just decrease total Foxc signaling sufficiently but not necessarily indicate distinct functions of the two genes as proposed. The question is whether *Foxc1* deletion would affect the phenotype of *Foxc2* mutants if *Foxc2* is efficiently deleted. Why is *Foxc1* staining apparent on the cell membrane and present in the *Foxc1* mutant in Figure 3—figure supplement 1?

2) The conclusion 'Cell elongation and junctional integrity is markedly impaired in compound EC- specific *Foxc1*; *Foxc2* mutants resulting in increased apoptosis in lymphatic collecting vessels' requires more supporting evidence. We see one image (Figure 3—figure supplement 3) aiming to support the claim that cell elongation and junctional integrity is impaired but it is not clear how representative this image is. It is surprising that such a high number of apoptotic cells is detected in control vessels, drawing into question the specificity of the caspase-3 staining, and whether the signal is indeed detected in the LECs. No evidence is provided for cell elongation defects leading to increased apoptosis. This statement should also be demonstrated: "Of note, apoptotic bodies were more frequently observed at branched areas of the lymphatic collecting vessels potentially indicating areas of valve degeneration." The authors should expand on the cellular mechanisms underlying the presumed rupture of lymphatic vessels and accumulation of chylous ascites in EC-*Foxc1*;*Foxc2*-DKO pups. Ultrastructural analyses could be informative here.

3) Inconsistent quality of staining between different experiments and even panels in the same figure (as if staining and imaging is done at different time points and not using littermate mice) make assessment of the phenotypes difficult, especially when a single high magnification image is provided as a supporting evidence. For example – Figure 3—figure supplement 2 VE-cadherin staining, Figure 2E-F CD31 staining. Figure 3 and Figure 3—figure supplement 3H show low PROX1 levels overall in *Foxc1*;*Foxc2* double mutants. Figure 3—figure supplement 3D shows instead extremely high PROX1 levels, along with 'discontinuous junctions', making it difficult to judge if this is a representative image. The junctional phenotype presented in Figure 3—figure supplement 3D is indeed dramatic (although the junctions do not seem to be 'discontinuous', but instead appear to have higher VE-cadherin levels), but the phenotype is seemingly different in Figure 3—figure supplement 2D where the phenotype is not obvious at all. 'Discontinuous junctions' in the *Foxc1*;*Foxc2* double mutants should be demonstrated more convincingly.

4) The authors should provide further evidence to demonstrate how relevant the described in vivo phenotypes are to those described in vitro. If *Foxc1* and *Foxc2* have distinct functions in regulating the actin cytoskeleton, it is not clear why *Foxc2* is capable of functionally substituting for *Foxc1* in vivo.

5) Previous studies have demonstrated that FOXC2 is regulated by oscillatory shear forces in LECs, and that such forces likely play a key role in valve development. In Figure 4 the authors use in vitro flow studies to conclude that FOXC1 expression is regulated primarily by laminar shear while FOXC2 primarily by oscillatory shear. However, the endothelial cells shown in Figure 4A do not appear aligned with flow as would be expected for laminar shear and the authors fail to identify a clear mechanism that would account for such a distinct regulation. This theory appears to rely primarily on antibody detection of nuclear FOXC1 which is not as robustly detected as FOXC2. In vivo studies, especially Figure 10, suggest that these transcription factors are functionally redundant at the protein level as replacement of FOXC1 with FOXC2 does not impair valve maturation. Much of the data shown are carefully analyzed, but these findings and their connection to where FOXC1 is best detected in vivo in the valve tip seems like an over-reach. More specific mechanistic data identifying how such remarkable distinctions in shear regulation are achieved should be provided.

6) Figure 1—figure supplement 1: What is the level of FOXC2 protein present in LEC of EC-*Foxc2*-KO compared to mRNA? *Foxc2* mRNA appears to be reduced by approximately 50% compared to control EC *Foxc2* levels. Does this level of reduction in EC-*Foxc2*-KO animals explain the partial penetrance of the phenotype of chylous ascites?

7) Figure 3—figure supplement 2: A lower power image depicting the broader profile of α9-integrin expression throughout multiple valves would enable the reader to determine how α9-integrin deposition is affected more globally in EC-*Foxc2*-KO compared to EC-*Foxc1*;*Foxc2*-DKO mice.

8) Figure 6: How do the changes in Prickle1, Arhgap21 and Arhgap23 levels compare to other gene expression changes in LEC-*Foxc1*;*Foxc2*-DKO LEC? It would be informative to provide more detail as to why these three genes were selected for further analysis. Were they the only genes involved in RhoA/ROCK signalling that were changed in expression? It would also be informative to know whether RAS/ERK signalling is affected in mutant LEC at this stage of development, given the authors' previous work which demonstrated regulation of this key pathway by FOXC1 and FOXC2 in embryonic LEC.

9) Figure 6D: Additional controls are required in which ChIP of both FOXC1 antibodies at a negative control region of the genome (e.g. an enhancer region demonstrated or predicted not to bind FOXC1) is assessed. This should demonstrate the selectivity of FOXC1 antibodies in terms of their specific versus non-specific/background binding to DNA and may help to explain the distinct results observed with each antibody in different regions of the genome.

10) Figure 10D-F: It is not clear from the images presented that a call can be made regarding FOXC2 levels being modestly increased in *Foxc1*^c2/c2^ mice. What is the remaining signal in FOXC1 stained *Foxc1^c2/c2^* tissue?

---

## [Author Response]

Essential revisions:1) The phenotype caused by early postnatal deletion of Foxc1 seems modest, and is characterized by a larger proportion of immature valves in mesenteric lymphatic vessels. It thus becomes important to know how the authors define immature valves. It is not obvious why those shown in Figure 2E, F are defined as immature, and not representing valves that are imaged from a different (90 degrees) angle compared to valves that show the typical v-shape. Staining of the valve leaflets would demonstrate this more clearly. The inclusion of staining for markers of valve maturation such as α5-laminin and/or α9-integrin would help to demonstrate this point. To more convincingly demonstrate that Foxc1 modulates the Foxc2 KO phenotype, side-by-side comparison of Foxc2 single and Foxc1;Foxc2 double mutants should be shown. Of note, deletion of Foxc2 seems to be incomplete (Figure 1—figure supplement 1). Considering that the authors show that Foxc2 can compensate for Foxc1 function, Foxc1 deletion in combination with the incomplete Foxc2 deletion may just decrease total Foxc signaling sufficiently but not necessarily indicate distinct functions of the two genes as proposed. The question is whether Foxc1 deletion would affect the phenotype of Foxc2 mutants if Foxc2 is efficiently deleted. Why is Foxc1 staining apparent on the cell membrane and present in the Foxc1 mutant in Figure 3—figure supplement 1?

The authors thank the reviewers for this comment and careful assessment of our described phenotypes in our *Foxc1*, *Foxc2*, and compound *Foxc1; Foxc2* mutant mice. To address the first part of this comment, we have included new representative images of α9-integrin/VE-Cadherin immunostaining of lymphatic valve regions side-by-side with PROX1/CD31 immunostained lymphatic valves in Figure 2. Inset images of only the α9-integrin channel more clearly show the presence of mature, intraluminal bi-leaflet valve structures in Control and EC-*Foxc1*-KO mice as well as valve regions that are considered immature that are not characterized by a distinct intraluminal leaflet structure by α9-integrin immunostaining. In EC-*Foxc2*-KO mice, we have observed that overall, α9-integrin expression is reduced compared to littermate control mice. However, there are regions characterized by the presence of intraluminal leaflets, as depicted by PROX1, CD31, and VE-Cadherin immunostaining, that are shortened in length. We describe these observations in the subsection “FOXC1 and FOXC2 are required for postnatal lymphatic valve maturation and maintenance”.

To address the second part of this comment, we have also updated Figure 2 to include representative images showing side-by-side comparison of EC-*Foxc2*-KO mice and EC-*Foxc1; Foxc2*-DKO mice. Figure 2H and N depict representative images of a mesentery collecting vessel immunostained with PROX1/CD31 and Figure 2R and T depict 10X power images of collecting vessels immunostained with PROX1 to show the difference in phenotypes more clearly.

To address the third part of this comment, we performed immunostaining of FOXC1, FOXC2, and VEGFR-3 in mesentery collecting vessels of EC-*Foxc1*-KO, EC-*Foxc2*-KO, EC-*Foxc1; Foxc2*-DKO mice and respective littermate controls and included representative images in Figure 2—figure supplement 1. In Figure 2—figure supplement 1G, we show that FOXC2 expression is markedly reduced in EC-*Foxc2*-KO mice and FOXC1 expression is reduced compared to littermate controls, likely as a result of perturbed flow contributing to the reduced induction of FOXC1 expression. In contrast, Figure 2—figure supplement 1E shows strong reduction of FOXC1 expression but no discernible difference in FOXC2 expression. Thus, the maintenance of FOXC2 expression in EC-*Foxc1*-KO mice may in part explain the differences in phenotype severity observed in *Foxc2* mutant mice compared to *Foxc1* mutant mice. We have revised our Discussion section (second paragraph) to note this important difference. The difference between our qPCR data, depicted in Figure 2—figure supplement 1B, and our immunostaining evidence is likely attributable to the fact that CD31-positive cells were isolated from cardiac tissue to use for qPCR analysis. It is likely that FOXC2 expression levels are different between lymphatic endothelium in mesentery collecting vessels and the cardiac vasculature. Thus, this may explain the differences in deletion efficiency. The repeated immunostaining of collecting vessels with FOXC1, FOXC2, and VEGFR-3 antibodies also now shows clear expression of FOXC1 in the nucleus of both lymphatic endothelial cells and smooth muscle cells of arterioles. As shown in Figure 2—figure supplement 1E, G, and I, there is no membrane signal detected. We suspect that there may be some cross-reactivity with our secondary antibodies used for the immunostaining combination of FOXC1 and VE-Cadherin, thus warranting our use of VEGFR-3 antibody to address this issue and verify the reduction of FOXC1 expression in the endothelium of collecting vessels.

2) The conclusion 'Cell elongation and junctional integrity is markedly impaired in compound EC- specific Foxc1; Foxc2 mutants resulting in increased apoptosis in lymphatic collecting vessels' requires more supporting evidence. We see one image (Figure 3—figure supplement 3) aiming to support the claim that cell elongation and junctional integrity is impaired but it is not clear how representative this image is. It is surprising that such a high number of apoptotic cells is detected in control vessels, drawing into question the specificity of the caspase-3 staining, and whether the signal is indeed detected in the LECs. No evidence is provided for cell elongation defects leading to increased apoptosis. This statement should also be demonstrated: "Of note, apoptotic bodies were more frequently observed at branched areas of the lymphatic collecting vessels potentially indicating areas of valve degeneration." The authors should expand on the cellular mechanisms underlying the presumed rupture of lymphatic vessels and accumulation of chylous ascites in EC-Foxc1;Foxc2-DKO pups. Ultrastructural analyses could be informative here.

We thank the reviewers for this helpful comment and careful critique of our data. We have revised our conclusion to more accurately state, ‘Cell elongation and junctional integrity is markedly impaired in compound EC-specific *Foxc1; Foxc2* mutants and is accompanied by increased apoptosis throughout lymphatic collecting vessels’ as the previous statement was not accurate. Our group previously reported that valves of LEC-*Foxc2-*KO mutant mice were characterized by increased apoptosis and apoptotic cells were often arranged in doublets with symmetrically organized PROX1-high apoptotic bodies, suggesting that apoptosis may occur in dividing cells. Additional evidence identified that half of the dying cells in LEC-*Foxc2*-KO mice were PROX1/Ki67-positive, demonstrating that valves of LEC-*Foxc2*-KO mice were characterized by abnormal activation of cell proliferation and increased cell death. This phenotype was associated with improper activation of TAZ signaling in the absence of FOXC2 (Sabine et al., 2015). We have included a summary of this finding in the subsection “Cell elongation and junctional integrity is markedly impaired in compound EC-specific *Foxc1*; *Foxc2* mutants and is accompanied by increased apoptosis throughout lymphatic collecting vessels”. We then sought to characterize junctional integrity and apoptosis in compound *Foxc1; Foxc2* mutant mice to assess if there was a synergistic effect of inactivation of both *Foxc1* and *Foxc2* in the lymphatic endothelium.

To further address this reviewer comment, we carefully re-evaluated our caspase-3 immunostaining and repeated analysis to characterize the percent of PROX1/cleaved caspase-3 positive lymphatic endothelial cells in 20X high-power field images. Figure 2—figure supplement 4I-K has been updated to include new representative images and quantitative analysis. In addition to representative images of maximum intensity projections of PROX1/cleaved caspase-3 immunostained collecting vessels, we also show panels of z-sections from selected regions in higher magnification where PROX1/cleaved caspase-3 positive cells were identified. We also now show images from two compound *Foxc1; Foxc2* mutant littermates that show apoptotic bodies are in closer proximity to branched regions.

3) Inconsistent quality of staining between different experiments and even panels in the same figure (as if staining and imaging is done at different time points and not using littermate mice) make assessment of the phenotypes difficult, especially when a single high magnification image is provided as a supporting evidence. For example – Figure 3—figure supplement 2 VE-cadherin staining, Figure 2E-F CD31 staining. Figure 3 and Figure 3—figure supplement 3H show low PROX1 levels overall in Foxc1;Foxc2 double mutants. Figure 3—figure supplement 3D shows instead extremely high PROX1 levels, along with 'discontinuous junctions', making it difficult to judge if this is a representative image. The junctional phenotype presented in Figure 3—figure supplement 3D is indeed dramatic (although the junctions do not seem to be 'discontinuous', but instead appear to have higher VE-cadherin levels), but the phenotype is seemingly different in Figure 3—figure supplement 2D where the phenotype is not obvious at all. 'Discontinuous junctions' in the Foxc1;Foxc2 double mutants should be demonstrated more convincingly.

We thank the reviewers for this helpful comment and careful critique of our representative images. To address the first part of the reviewer comment, we have repeated immunostaining of tissues and re-evaluated the post-processing of our representative images for several figures to address issues with inconsistencies. This includes new images for PROX1/CD31 immunostaining in Figure 2C, E, I, and K and new images for PROX1/VE-Cadherin immunostaining in Figure 2—figure supplement 4A-H that now show consistent PROX1 expression levels with representative images in Figure 2. Additionally, our new representative images for Figure 2—figure supplement 3E-H and Figure 2—figure supplement 4A-H show consistent VE-Cadherin expression levels among different tissue samples. Furthermore, our new images of VE-Cadherin immunostaining in EC-*Foxc1; Foxc2*-DKO mice more clearly depict that collecting vessels in these mice are characterized not only by discontinuous junctions, but the increased presence of overlapping junctions as well. We have also generated videos of 3D reconstructions of the collecting vessels depicted in Figure 2—figure supplement 4A-H (Videos 1 – 4), characterizing this observation.

4) The authors should provide further evidence to demonstrate how relevant the described in vivo phenotypes are to those described in vitro. If Foxc1 and Foxc2 have distinct functions in regulating the actin cytoskeleton, it is not clear why Foxc2 is capable of functionally substituting for Foxc1 in vivo.

We have now included new representative images of VE-Cadherin immunostaining in EC-*Foxc2*-KO and EC-*Foxc1; Foxc2*-DKO mutants in Figure 2—figure supplement 4A-H that shows differences in the presence of discontinuous junctions in collecting vessels of EC-*Foxc2*-KO mice, but both discontinuous and overlapping junctions present in collecting vessels of EC-*Foxc1; Foxc2*-DKO mice. This new data recapitulates our in vitro evidence depicting the same observations in *FOXC2*-KD and *FOXC1; FOXC2-*KD cultured lymphatic endothelial cells as represented in Figure 4—figure supplement 1, thus providing additional support for the relevancy of our in vivo and in vitro observations. In regard to the second part of the reviewer comment, although FOXC1 and FOXC2 are functionally similar, as characterized by their similar DNA-binding capacity, their transcriptional targets are likely to be different under oscillatory or laminar shear stress. For example, our group previously reported that loss of *FOXC2* leads to distinct phenotypes in lymphatic endothelial cells under oscillatory shear stress and laminar shear stress (Sabine et al., 2012, Supplementary Figure 5A). Thus, as FOXC1 is induced only by laminar shear, but FOXC2 is induced by both laminar and oscillatory shear, their downstream targets are likely different within the collecting vessels of lymphatic endothelial cells. We have also updated the fourth paragraph of the Discussion section to elaborate on these differences.

5) Previous studies have demonstrated that FOXC2 is regulated by oscillatory shear forces in LECs, and that such forces likely play a key role in valve development. In Figure 4 the authors use in vitro flow studies to conclude that FOXC1 expression is regulated primarily by laminar shear while FOXC2 primarily by oscillatory shear. However, the endothelial cells shown in Figure 4A do not appear aligned with flow as would be expected for laminar shear and the authors fail to identify a clear mechanism that would account for such a distinct regulation. This theory appears to rely primarily on antibody detection of nuclear FOXC1 which is not as robustly detected as FOXC2. In vivo studies, especially Figure 10, suggest that these transcription factors are functionally redundant at the protein level as replacement of FOXC1 with FOXC2 does not impair valve maturation. Much of the data shown are carefully analyzed, but these findings and their connection to where FOXC1 is best detected in vivo in the valve tip seems like an over-reach. More specific mechanistic data identifying how such remarkable distinctions in shear regulation are achieved should be provided.

As indicated in the Materials and methods section, the flow experiment presented in Figure 3 (formerly Figure 4) was run for only 24 hours (subsection “Cell transfection and immunostaining”). This shorter time – 24-hour flow instead of 48-hour flow presented in our previous publications (Sabine et al., 2012, and Sabine et al., 2015) – is not sufficient to drive an important cell elongation in the direction of flow. In Author response image 1, we present to the reviewer a kinetic view of LSS-mediated cell elongation in the direction of flow (horizontal on the provided pictures) showing that, in our hands, at 24 hours cells are not much elongated in comparison to 48 hours (Author response image 1).

**Author response image 1. sa2fig1:** Kinetics of LEC elongation in the direction of flow under 4 dyn/cm^2^ laminar shear stress. Confluent LECs were seeded and 24 hours later cultivated for 1 hour, 6 hours, 24 hours or 48 hours either under static conditions or oscillatory shear stress (4 dyn/cm^2^, 1/4 Hz oscillations) or unidirectional laminar shear stress (4 dyn/cm^2^). The upper part of the figure shows representative images of LECs stained for VE-cadherin (grey). Scale bar, 50 µm. The lower part of the figure shows graphs with individual cell measurement of cell shape (length:width ratio) and alignment to flow axis (horizontal) for each time point. LECs are defined as elongated when they have a length:width ratio greater than 2 are aligned to flow with an angle lower than 45°. These cells are highlighted in the red box and the percentage of total population is indicated in red on the figure.

In addition, we also provide the reviewer with the quantification of length:width ratio for static, LSS and OSS conditions presented in Figure 3, which shows that there is a significant elongation of cells under LSS in average (Author response image 2). To clarify this point, we have now indicated in the legend of Figure 3 as: “Representative images of cultured LECs under static, OSS, or LSS show increased expression of FOXC1 when subjected to 24 hours to LSS, whereas FOXC2 is induced by both OSS and LSS.”

The observation that FOXC1 and FOXC2 expression are already different in LECs after only 24 hours under laminar shear stress is interesting as it suggests that FOXC1 upregulation by laminar flow is an early response of LECs to directional flow. How expression levels of FOXC1 and FOXC2 are distinctly regulated by the directionality of flow is a very interesting question that addresses the more general one of how LECs sense flow direction and are able to respond differently to LSS and OSS. Answering such a question would require further investigation and, in our opinion, is beyond the scope of this paper in which the central message is that each factor is differentially regulated by flow and plays slightly distinct but complementary roles in regulating the cell cytoskeleton in response to shear stress.

**Author response image 2. sa2fig2:** Quantification of cell elongation in the direction of flow corresponding to Figure 3. Graphs show individual cell measurement of cell shape (length:width ratio) and alignment to flow axis (horizontal) for each time point. LECs are defined as elongated when they have a length:width ratio greater than 2 are aligned to flow with an angle lower than 45°. These cells are highlighted in the red box and the percentage of total population is indicated in red on the figure.

6) Figure 1—figure supplement 1: What is the level of FOXC2 protein present in LEC of EC-Foxc2-KO compared to mRNA? Foxc2 mRNA appears to be reduced by approximately 50% compared to control EC Foxc2 levels. Does this level of reduction in EC-Foxc2-KO animals explain the partial penetrance of the phenotype of chylous ascites?

As previously described in our response to reviewer comment #1, Figure 2—figure supplement 1F and G show a marked reduction of FOXC2 protein within collecting vessels of EC-*Foxc2*-KO mice compared to littermate controls. We also suggested that the differences observed in reduction of mRNA levels compared to what we observe in regard to protein expression may be attributable to the different vascular beds analyzed. The difference in the penetrance of the phenotype of chylous ascites in the current study may be attributable to both the timeline of experimental analysis as well as the differences in Cre-drivers used. In our group’s previous study, tamoxifen was administered to *Prox1-Cre^ERT2^; Foxc2^fl/fl^* mice beginning at P4 with approximately 50% of mutants developing chylous ascites 4 days after tamoxifen administration and nearly all mutants presenting chylous ascites 6-8 days after tamoxifen administration (Sabine et al., 2015, Figure 7A). In the current study, tamoxifen administration started at P1 and we primarily investigated tissues at P6. We believe it is likely that our *Cdh5-Cre^ERT2^; Foxc2^fl/fl^* mice would present chylous ascites at later time points.

7) Figure 3—figure supplement 2: A lower power image depicting the broader profile of α9-integrin expression throughout multiple valves would enable the reader to determine how α9-integrin deposition is affected more globally in EC-Foxc2-KO compared to EC-Foxc1;Foxc2-DKO mice.

We thank the reviewers for this suggestion. Figure 2—figure supplement 3A-D now depicts 10X low power images of α9-integrin/VE-Cadherin immunostaining to show that a few, α9-integrin-positive degenerating valve regions are detected in collecting vessels of EC-*Foxc2*-KO mice at P6, but the collecting vessels of EC-*Foxc1; Foxc2*-DKO mice are absent of α9-integrin-positive valve regions.

8) Figure 6: How do the changes in Prickle1, Arhgap21 and Arhgap23 levels compare to other gene expression changes in LEC-Foxc1;Foxc2-DKO LEC? It would be informative to provide more detail as to why these three genes were selected for further analysis. Were they the only genes involved in RhoA/ROCK signalling that were changed in expression? It would also be informative to know whether RAS/ERK signalling is affected in mutant LEC at this stage of development, given the authors' previous work which demonstrated regulation of this key pathway by FOXC1 and FOXC2 in embryonic LEC.

We agree with the reviewers that providing additional details regarding the choice to focus on Prickle1, Arhgap21, and Arhgap23 would be helpful. We have now updated Figure 5—figure supplement 1 in the revised manuscript to include additional data from our RNA-seq analysis for LECs isolated from the dorsal skin of embryonic LEC-*Foxc1; Foxc2*-DKO mice and littermate controls in panel K. Figure 5—figure supplement 1K now shows the changes in expression for *RhoA, Rock1,* and *Rock2*, and several GTPase activating proteins associated with regulation of RhoA signaling and endothelial barrier function and lumen maintenance (van Buul, Geerts, and Huveneers, 2014, Barry et al., 2016). Here, we see that there is a modest, yet significant increase in *Arhgap18* expression and a significant reduction of *Arghap20*, although its expression was generally lower compared to other GAPs. Because of the previously reported evidence of the formation of a physical complex between Prickle1 and Arhgap21/23 (Zhang et al., 2016) and our observation that all three genes were significantly downregulated in our RNA-seq analysis, we focused our investigation on these putative downstream targets. In regard to changes in RAS/ERK signaling, we attempted to perform whole-mount immunostaining of pERK using the same antibody utilized in our group’s previous study (Cell Signaling #4370T) but we were not successful in identifying a positive signal in control or mutant lymphatic valves at P6, testing the antibody at various concentrations. While we are interested in characterizing whether RAS-ERK signaling may be affected in our mutants during postnatal development in other lymphatic vascular beds, the authors believe that this is not of focus for the mechanism described in this study and is beyond its scope.

9) Figure 6D: Additional controls are required in which ChIP of both FOXC1 antibodies at a negative control region of the genome (e.g. an enhancer region demonstrated or predicted not to bind FOXC1) is assessed. This should demonstrate the selectivity of FOXC1 antibodies in terms of their specific versus non-specific/background binding to DNA and may help to explain the distinct results observed with each antibody in different regions of the genome.

We thank the reviewers for this important suggestion. We have now included Figure 5—figure supplement 1I and J to directly address this comment. Panel I depicts the location of primers used to amplify a transcriptionally active region of the *ICAM1* promoter that is not predicted to bind to FOX transcription factors. Panel J then shows data from three separate experiments showing no detection of band signals from ChIP with our FOXC1 or FOXC2 antibodies. Within the text of the manuscript, we have also included relevant literature that has identified variations in immunohistochemistry related work for antibodies recognizing the same immunogen peptide that are dependent on the source of the manufacturer. This information is included in the second paragraph of the subsection “FOXC1 and FOXC2 regulate LEC expression of negative RhoA signaling regulators PRICKLE1, ARHGAP21, and ARHGAP23”.

10) Figure 10D-F: It is not clear from the images presented that a call can be made regarding FOXC2 levels being modestly increased in Foxc1^c2/c2^ mice. What is the remaining signal in FOXC1 stained Foxc1^c2/c2^ tissue?

We agree with the reviewers that it is likely not accurate to describe that FOXC2 levels are modestly increased. Thus, this statement has now been removed from the revised manuscript. To verify that FOXC1 protein expression is absent in collecting vessels of *Foxc1^c2/c2^* mice, we have also included new representative images of FOXC1/VEGFR-3 immunostaining in Figure 9D and E. As previously stated in our reply to reviewer comment #1, we suspect that the remaining signal in the FOXC1 channel of Figure 9H is non-specific and may be related to a cross-reactivity issue of secondary antibodies. However, we believe that Figures 9F-H clearly depict a reduction in nuclear FOXC1 expression as expected.